# Drug Resistance in Metastatic Breast Cancer: Tumor Targeted Nanomedicine to the Rescue

**DOI:** 10.3390/ijms22094673

**Published:** 2021-04-28

**Authors:** Vrinda Gote, Anantha Ram Nookala, Pradeep Kumar Bolla, Dhananjay Pal

**Affiliations:** 1Division of Pharmacology and Pharmaceutical Sciences, School of Pharmacy, University of Missouri-Kansas City, 2464 Charlotte Street, Kansas City, MO 64108, USA; vrindagote@mail.umkc.edu (V.G.); anfh3@mail.umkc.edu (A.R.N.); 2Department of Biomedical Engineering, College of Engineering, The University of Texas at El Paso, 500 W University Ave, El Paso, TX 79968, USA; pbolla@miners.utep.edu; 3Department of Basic Pharmaceutical Sciences, Fred Wilson School of Pharmacy, High Point University, High Point, NC 27268, USA

**Keywords:** theranotics nanoparticles, triple negative breast cancer, stimuli-responsive nanocarriers, breast cancer stem cells, siRNA, micro RNA

## Abstract

Breast cancer, specifically metastatic breast, is a leading cause of morbidity and mortality in women. This is mainly due to relapse and reoccurrence of tumor. The primary reason for cancer relapse is the development of multidrug resistance (MDR) hampering the treatment and prognosis. MDR can occur due to a multitude of molecular events, including increased expression of efflux transporters such as P-gp, BCRP, or MRP1; epithelial to mesenchymal transition; and resistance development in breast cancer stem cells. Excessive dose dumping in chemotherapy can cause intrinsic anti-cancer MDR to appear prior to chemotherapy and after the treatment. Hence, novel targeted nanomedicines encapsulating chemotherapeutics and gene therapy products may assist to overcome cancer drug resistance. Targeted nanomedicines offer innovative strategies to overcome the limitations of conventional chemotherapy while permitting enhanced selectivity to cancer cells. Targeted nanotheranostics permit targeted drug release, precise breast cancer diagnosis, and importantly, the ability to overcome MDR. The article discusses various nanomedicines designed to selectively target breast cancer, triple negative breast cancer, and breast cancer stem cells. In addition, the review discusses recent approaches, including combination nanoparticles (NPs), theranostic NPs, and stimuli sensitive or “smart” NPs. Recent innovations in microRNA NPs and personalized medicine NPs are also discussed. Future perspective research for complex targeted and multi-stage responsive nanomedicines for metastatic breast cancer is discussed.

## 1. Introduction

Breast cancer is the most frequently diagnosed cancer in women. In 2018, 2.1 million women were diagnosed with breast cancer, and approximately one new case was diagnosed every 18 s [1]. Breast cancer can be regarded as a heterogeneous disease as seen at the molecular level. Early breast cancer (when the cancer is confined in the breast or spreads to the axillary lymph nodes) is considered curable. Improvements in breast cancer therapeutics have led to increasing survival rate, which is more than 85% for 5 years in US. Advanced breast cancer, which has metastasized to various organs, may not completely be curable with currently available chemotherapeutics. The last decade has seen a rise of targeted therapies for advanced or metastatic breast cancer considering the heterogeneity of the disease. Recently, breast cancer therapies emphasize biologically-directed therapies, personalized treatment, and de-escalation of the chemotherapy and treatment for reducing the adverse effects of chemotherapeutics. Although the 5-year survival rate of advanced or metastasized breast cancer is low (28%), the main goal of targeted therapy is to prolong survival, control symptoms, and lower toxicity associated with cytotoxic drugs and, in turn, improve the quality of life [2].

### 1.1. Breast Cancer Classification

Breast cancer treatment is divided into two treatment options. This includes regional treatment and systemic therapy. The stage of the cancer and its type, based on molecular and histological classification, mainly decide the treatment options. Early breast cancer, in stage I and II, in which the tumor is confined to the breast tissue or infected local lymph nodes, can be cured by mastectomy and or radiation therapy. Breast cancer is commonly recognized by the presence or absence of three major receptors on the cell surface. These are hormone receptors such as estrogen receptor (ER), progesterone receptor (PR), and human epidermal growth factor-2 receptor (HER2). Metastatic breast cancer (MBC) is usually treated by chemotherapy, hormone therapy, and targeted therapy. Depending on the molecular nature of the cancer and its origin in the breast tissue, Perou and Sorlie reported the intrinsic classification of breast cancer in 2000 [3]. They classified breast cancer into four subtypes: (i) Luminal A; (ii) Luminal B, both expressing ER; (iii) basal-like breast cancer; and (iv) HER2 enriched breast cancer without ER and PR expression [3]. In clinical practice, the surrogate classification of breast cancer is commonly used. This classification includes five types of breast cancers classified according to varying molecular and histological differences in breast cancer (Figure 1). This intrinsic classification shifted clinical management of the disease from being based on tumor biology to a molecular targeted approach. Another way of classifying breast cancer is by the presence and absence of receptors on the cell surface. Hormone positive breast cancer consists of ER and or PR receptors, while HER2 positive breast cancer is enriched with HER2 receptors but lacks ER and PR receptor expression. HER2 breast cancer cannot be treated with hormone therapy. Triple negative breast cancer (TNBC) lacks the expression of ER, PR, and HER2. This type of breast cancer cannot be treated with hormone therapy or HER2 targeted therapy. Chemotherapy is the only current option for the treatment of TNBC [2].

### 1.2. Breast Cancer Pathophysiology and Metastasis

The exact mechanism of breast cancer initiation is currently not fully delineated. However, efforts have been made to understand breast cancer formation and progression at the molecular level. Various models or theories are proposed for this. The cancer stem cell model assumes that stimulation of precursor cancer stem cells triggers initiation and progression of breast cancer. This theory also posits that cellular diversity in breast cancer and tumor hierarchy are generated by the cancer stem cells [4]. The clonal evolution model postulates that random mutation, along with clonal selection at the genetic level, give rise to the cancer cellular heterogeneity commonly observed in breast cancer [5,6]. These two models of breast cancer evolution can be interconnected by the fact that breast cancer stem cells may evolve in a clonal evolution [7]. Molecular progression of breast cancer rises along two divergent pathways. These are the low-grade pathway and high-grade pathway. The low-grade pathway is characterized by changes in the gene expression for the majority of the genes belonging to the ER phenotype and diploid or near diploid karyotypes. The Luminal A group of breast cancer, and to some extent, Luminal B, fall into this pathway. On the other hand, the high-grade pathway is characterized by gene alterations, including loss of the 13q gene, gain of a chromosomal region-11q13, and amplification of 17q12 gene. The 17q12 gene codes for HER2 in breast cancer cells. There is a high level of gene expressions involved in cellular proliferation and cell cycle maintenance in metastatic breast cancer [8]. HER2 positive and TNBC progress by the high-grade pathway of cancer progression [9]. A number of genes become either mutated, amplified, or both in various types of breast cancer. Some examples of these genes are: PIK3CA (altered in 30% of tumors), PTEN (altered in 16% of tumors), TP53 (altered in 41% of tumors), MYC (altered in 20% of tumors), CCND1 (altered in 16% of tumors), ERBB2 (altered in 13% of tumors), and GATA3 (altered in 10% of tumors), as reported in patients with early breast cancer [10]. These genes regulate important modulators of cell cycles. Breast cancer, similar to many other cancers, represses the p53 gene, and genes for increased production of cyclin D1 get activated. In addition, in breast cancer, pathways such as HER2, MYC, and FGFR1, responsible for inhibition of apoptosis and sustained proliferation of cancer cells, endure activation [11].

Advanced breast cancer can metastasize into various organs, including the auxiliary lymph nodes, lungs, bones, brain, liver, and peritoneal cavity. The most common metastatic site for breast cancer is the bones. Approximately 67% of advanced breast cancer tumors metastasize in the bones. Luminal B (79%) and Luminal A (70%) types of breast cancer have the highest percentages of metastasis to the bones, while HER2+ and TNBC or basal-like breast cancer have 60% and 40% probability of infecting the bone tissue. The other most common site for breast cancer metastasis are the liver, auxiliary lymph nodes, and the lungs. Almost 37% of advanced breast cancer can metastasize to the liver and lungs, respectively, while the metastasis of breast cancer to the auxiliary lymph nodes varies between 30–50%. Liver metastasis of TNBC and HER2+ is more frequent compared to the Luminal breast cancer subtype. HER2+ has a 45% probability and TNBC has a 35% probability of metastasizing to the liver tissue. All breast cancer subtypes except TNBC have a higher chance for angiogenesis and infecting the auxiliary lymph nodes. However, for the lungs, Luminal A and B have a slightly lower metastasis (25–30%), while TNBC and HER2+ have a slightly higher chance of metastasis (45–35%) to the lungs. Metastasis of breast cancer into vital organs, such as the lungs, liver, and brain, can severely reduce the survival period of patients. Application of nanomedicines can definitely aid in reducing the burden by breast cancer metastasis. Approximately 12.6% of the cancer metastasizes to the brain tissue. In this particular metastatic site, TNBC and HER2+ (25–30%) have a higher level of metastasis compared to Luminal A and Luminal B (5–15%) types of breast cancer. Other less common sites of metastasis are mammary internal chain lymph nodes (10–40%), contralateral breast (6%), and supraclavicular lymph nodes (1–4%). Figure 2 depicts various metastatic sites in breast cancer [2].

## 2. Molecular Mechanism for Breast Cancer Resistance, Metastasis, and Relapse

### 2.1. Increased Drug Efflux

An elevated efflux of chemotherapeutic drugs from the cancer cells leads to lower drug accumulation. This is the leading cause of drug resistance to chemotherapeutic drugs [12,13,14,15]. Drug efflux transporters, also called efflux pumps, are mainly responsible for the development of multidrug resistance (MDR) in cancer cells [16,17,18,19]. Efflux transporters belong to the ATP-binding cassette (ABC) superfamily of transporters, responsible for transport of solutes in and out of the cell membrane through energy derived from ATP hydrolysis. The human genome contains 48 ABC genes classified into seven subfamilies (ABCA-ABCG). Among these are ABCB1, ABCC1, and ABCG2. These transporters are greatly involved in the acquisition of MDR to cancer chemotherapeutics [20,21,22]. Efflux transporters mainly include: (i) P-gp, an ABC sub-family-B member-1 encoded in the human body by ABCB1 gene; (ii) BCRP, a member of the white sub-family and ABCG member 2 encoded in the human body by the ABCG2 gene; and (iii) multidrug resistance associated protein-1 (MRP-1), an ATP-binding cassette sub-family C member 2 encoded in the human body by the ABCC2 gene [23,24]. Figure 3 depicts various mechanisms of drug resistance, including efflux pump-mediated mechanisms of MDR and efflux pump-independent drug resistance mechanisms [25].

#### 2.1.1. P-Glycoprotein

P-glycoprotein (P-gp) is also known as multidrug resistance protein-1 (MDR-1) or cluster of differentiation 243 (CD243). It is also the first member of the ABC superfamily and is responsible for efflux of xenobiotics [17,26]. P-gp is composed of two homologous halves with 1280 amino acids. Each half consists of six hydrophobic transmembrane domains (TMD) with an ATP binding site or nucleotide-binding domain (NBD) that hydrolyzes ATP. ATP binding causes conformational changes in the transporter, essential for the functioning of the transporter [27]. Over-expression of P-gp in cancer cells results in MDR. P-gp has multiple drug binding sites on its transporter structure that bind to a variety of chemotherapeutic drugs, including doxorubicin, etoposide, paclitaxel, and many others [28,29,30,31,32]. P-gp expression varies in various types of cancers. Colon, pancreas, liver, adrenal gland, and kidney cancers demonstrate highest levels of P-gp expression, while intermediate P-gp expression is seen in soft tissue carcinomas, neuroblastoma, and hematological malignancies. Breast, ovary, lung, and esophageal cancers initially display low P-gp levels. But the levels of P-gp efflux transporters increases after the cancer shows resistance to the chemotherapeutic treatment [33,34].

P-gp inhibitors can be divided into first, second, and third generation drugs, which primarily block P-gp and aid reversal of MDR. First generation drugs appear less potent, non-selective, and have a low P-gp binding affinity. High doses of these inhibitors are required to reverse MDR. This can result in toxic side effects for the patients. Trifluoperazine, Cyclosporine-A, Verapamil, Quinidine and Reserpine, Tamoxifen, Vincristine, and Yohimbine are examples of first generation P-gp inhibitors [35,36]. The inadequacies of pharmacokinetic and pharmacodynamics of first-generation P-gp inhibitors paved the way for the development of second generation P-gp inhibitor drugs. Similar to the first generation agents, these agents inhibit metabolism and excretion of chemotherapeutic drugs by blocking effects of P-gp. Structural modification of first-generation P-gp inhibitors helped in achieving a better pharmacological response, better tolerability, and reduced toxicity. Drugs such as Valspodar (PSC 833), Dexverapamil, Biricodar citrate (VX-710), and Dexniguldipine are second-generation P-gp inhibitors [36]. Some shortcomings of second-generation P-gp inhibitors, such as interaction with cytochrome P450 3A4 affecting drug pharmacokinetics were overcome in third-generation P-gp inhibitors. Zosuquidar-LY335979, Elacridar-GF120918, Annamycin, and Mitotane (NSC-38721) are examples of third-generation P-gp inhibitors [23]. Tariquidar can be regarded as an ideal P-gp inhibitor since it has demonstrated P-gp inhibition activity in cancer chemotherapy clinical trials [37,38]. The drug was also effective in reversing MDR in chemotherapy resistant advanced breast cancer in Phase II clinical trials [39,40], NCT00048633. Tariquidar can be used to overcome MDR in breast cancer. Despite some clinical impact, these third-generation P-gp inhibitors cause high toxicity. Zhong et al. demonstrated that co-delivery of folic acid-targeted nano-erythrocyte can be utilized to overcome MDR in breast cancer in vivo [41].

#### 2.1.2. Breast Cancer Resistance Protein

Breast cancer resistance protein (BCRP) was first identified in a drug-resistant human breast cancer cell line that was treated simultaneously with mitoxantrone and Tariquidar, which are P-gp inhibitors [34,42,43]. BCRP is coded by the ABCG2 gene, and it belongs to the ABCG subfamily of ABC transporters. BCRP is a half-transporter composed of N-terminal NBD and a carboxy-terminal TMD containing six membrane-spanning helices [44,45]. The transporter plays an important role in intercellular drug absorption, metabolism, and excretion, as well as toxicity. It also functions as an efflux pump for transport of anti-cancer drugs from the breast cancer cells to the extracellular environment, thus granting MDR to the tumor cells. Although this transporter was initially identified in breast cancer, it was later seen to cause MDR in most of the types of cancers [46]. Anthracyclines can develop resistance due to BCRP expressed on breast cancer cells [42,47]. In addition to cell membranes, BCRP is also expressed in intracellular vesicles. These vesicles generally retain drugs, but BCRP pumps the drugs out quickly. This is another reason for increased drug resistance due to BCRP efflux transporter [48]. BCRP is highly expressed in side-population cells in breast cancer. These cells possess stem cell-like properties and are mostly resistant to chemotherapy. Wiese et al. discusses BCRP/ABCG2 inhibitor patents and how these inhibitors can have additional benefits besides counteracting MDR. Among many inhibitors, the most promising ones are bivalent flavonoids, which have shown broad-spectrum inhibitory activity as compared to other classes of compounds. Bivalent flavonoids are also selective inhibitors of BCRP/ABCG2 [49].

#### 2.1.3. Multidrug Resistance Protein

Multidrug resistance protein (MRP) is a member of the human ABC family of cell membrane transporters known to cause MDR. This transporter was first discovered while studying the H69AR cell line, which is a small cell lung cancer drug-resistant line [50,51]. MRP1 transporter consists of 17 TMD spanning across the cell membrane. MRP is responsible for the transport of endogenous substances and xenobiotic drugs. MRP1 has been extensively studied over the last two decades for its role in developing drug resistance in various cancers. MRP1 is a 190 kDa protein with a P-gp transporter like core and an additional N-terminal TMD [52,53]. A distinct feature of MRP1 is that it is a basolateral transporter. This implies that MRP1 activity results in the movement of compounds into cells that lie below the basement membrane. The transporter prevents drug absorption from basolateral side and clears the drugs out of cells [54]. MRP1 can transport glutathione (GSH), chemotherapeutic drugs, and GSH-conjugated compounds [55]. This implies that the mechanism of transport of MRP1 is different from that of P-gp transport [55]. MRP1 is ubiquitously expressed in most of the human tissues. Thus, it is present in most of the tumors, including breast cancer, and plays an important role in MDR [56]. MRP1 is responsible for developing resistance to drug classes, such as vinca alkaloids, anthracyclines, camptothecins, epipodo-phyllotoxins, platinum compounds, nucleoside and nucleotide analogs, folate antimetabolites, and methotrexate drugs. MRP1 does not confer resistance to all taxenes except methotrexate. Breast cancer relapse is highly linked with increase in MRP1 activity in the tumor cells [54,57]. Resistance due to MRP/ABCC members (MRPs 1–3) is often caused by an increased efflux and leads to decreased intracellular accumulation of anti-cancer drugs. Drug targeting of MRP transporters can help to overcome resistance associated with breast cancer cells [58].

#### 2.1.4. Lung Resistance Protein

Lung resistance protein (LRP) is another transmembrane protein, which is encoded by the LRP gene. It was first discovered in non-small cell lung cancer cell line SW-1573 [59]. The protein is found in the cytoplasm and in the nuclear membrane of tumor cells. It is not a member of the ABC superfamily of transporter proteins [60]. Primary sequence analysis of LRP cDNA discovered that the amino acid sequence of LRP was 87.7% homologous to the brown rat vault protein, also called the major vault protein. Vaults are organelles broadly distributed and conserved in diverse eukaryotic cells. These organelles localize in nuclear pore complexes and form the central plug of the nuclear complexes. These proteins help in transport of substances in and out of the nuclear membrane. These vaults may play a role in MDR by regulating the nucleo-cytoplasmic movement of drugs [61]. LRP protein is overexpressed in most cancers, which results in lower accumulation of anti-cancer drugs in the nucleus [62]. Wood et al. observed that LRP concentration in saliva of breast cancer patients was significantly higher compared to healthy women [63]. Similar to P-gp, BCRP, and MDRP, LRP also causes resistance to compounds including alkaloids, anthracyclines, and epipodophyllotoxin. In addition to this, LRP also causes resistance to cisplatin and several atypical MDR drugs [61,64].

### 2.2. Breast Cancer Stem Cells

Breast cancer stem cells (BCSCs) represent a small subpopulation of breast cancer cells that play important roles in cancer progression, metastasis, and recurrence. BCSCs differ from other breast cancer cells in their ability to resist treatment with the current chemotherapeutic agents and radiotherapy [65]. Baseline BCSCs are inversely correlated with metastatic breast cancer treatment rate and survival [66]. A growing body of research points out to the expression of different surface markers and signaling molecules that can be explored as possible therapeutic targets. The ability of BCSCs to promote tumor formation is characterized by a high expression of surface marker CD44 and a low or complete absence of surface marker CD24 (CD44^+^/CD24^−/low^) [67]. Furthermore, the potential of BCSCs to metastasize is enhanced by the expression of an epithelial cell adhesion molecule that aids in cell adhesion, proliferation, and differentiation [68,69]. Independent of CD44^+^/CD24^−/low^ expression, BCSCs with increased aldehyde dehydrogenase 1 (ALDH1) activity have been shown to exhibit tumorigenic potential. Cells that exhibit CD44^+^/CD24^−/low^ and ALDH1 phenotype have greater tumorigenic potential compared to the cells expressing either one of the phenotypes. Cell numbers as low as 20 were enough to promote tumor formation when injected into humanized cleared fat-pads of non-obese diabetic (NOD)/severe combined immunodeficiency (SCID) mice [70]. Within the breast cancer tissue, BCSCs with CD44^+^/CD24^−/low^ phenotype are closely present to the invasive edge, whereas BCSCs with elevated ALDH1 activity are primarily found in the inner regions where hypoxia is predominant [71,72]. Current cytotoxic drugs have been shown to increase the BCSCs population by two mechanisms: (a) affecting only non-BCSCs and, thereby, increasing the proportion of BCSCs or (b) converting the non-BCSCs to BCSCs. For instance, paclitaxel, carboplatin, and 5-florouracil have been shown to be effective against non-BCSCs and increase the proportion of CD44^+^/CD24^−/low^ BCSCs in various cell lines [73,74]. BCSCs with CD44^+^/CD24^−/low^ and ALDH1 phenotypes isolated from different subtypes of breast cancer, based on molecular profiling of receptor expression (estrogen receptor, progesterone receptor, or human epidermal growth factor receptor 2), have similar gene expression profiles [72]. This suggests that a common therapeutic strategy can be employed to treat the BCSCs in different breast cancer subtypes.

Expression of various signaling molecules and transcription factors play an important role in BCSCs metastasis. Tribbles homolog 3 is a protein encoded by the TRIB3 gene. Its expression was directly correlated with poor breast cancer prognosis and BCSCs stemness by affecting the FOXO1-AKT-SOX2 pathway [75]. Activation of the Wnt/β-catenin pathway was shown in various subtypes of breast cancer [76]. Exposure to transforming growth factor-β1 (TGF-β1) increased the number of mammospheres in the in vitro model [77]. Several other signaling pathways are shown to be important for BCSCs, including activation of hedgehog, notch, nuclear factor kappa B (NF-κB), receptor tyrosine kinase, etc. [78]. Therefore, targeting BCSCs is an important determinant in the treatment of breast cancer. Several studies have been shown to affect the key signaling pathways and transcription factors to decrease the BCSCs population. For instance, VS-4718 and VS-6063 potently inhibited focal adhesion kinase (FAK) and, thereby, affected the Wnt/β-catenin pathway. Compared to paclitaxel alone, combination treatment with paclitaxel and VS-4718 or VS-6063 resulted in delayed tumor re-growth and metastasis in mice bearing tumors formed by patient-derived xenografts [79]. Quisinostat, a histone deacetylase inhibitor, in combination with doxorubicin, decreased the number of non-CSCs and BCSCs in a synergistic manner [80]. Plumbagin, a natural naphthoquinone derivative, was effective against BCSCs in endocrine resistant breast cancer by modulating the Wnt/β-catenin pathway [81]. Hydroxytyrosol affected the BCSCs sub-population with CD44^+^/CD24^−/low^ and ALDH1 activity by targeting the Wnt/β-catenin and TGF pathways (30460610). The combination of tamoxifen with napabucasin decreased stemness and rendered the cells sensitive to treatment by inhibiting signal transducers and activators of transcription 3 (STAT3) activation [82]. Iadademstat inhibited the formation of mammospheres induced by BCSCs in different breast cancer cell lines and patient derived xenografts [83].

The past two decades have seen advancements in the development of novel therapeutic formulations. Several research groups have utilized the approach of using a combination therapy to develop novel formulations to target BCSCs along with non-BCSCs and maintain the drugs for a longer time in circulation. For instance, the docetaxel and salinomycin combination was formulated in polylactide-co-glycolide/D-alpha-tocopherol polyethylene glycol 1000 succinate nanoparticles. This nanoparticle combination had a synergistic effect on the breast cancer cells and stem cells and were present in circulation for a longer time [84]. Zileuton loaded polymeric micelles were effective against BCSCs in vitro and in vivo [85].

### 2.3. Epithelial to Mesenchymal Transition (EMT)

BCSCs with different phenotypes, including the CD44^+^/CD24^−/low^ phenotype, have been shown to exhibit features of epithelial mesenchymal transition [77]. Epithelial mesenchymal transition (EMT) is a process that converts polarized epithelial cells into mesenchymal cells, which have an increased ability to migrate, produce extracellular matrix components, and metastasize [86]. The process involves activation of multiple signaling pathways and transcription factors [87]. BCSCs with the CD44^+^/CD24^−/low^ phenotype isolated from primary human breast cells showed a higher expression of vimentin, zinc finger E-box-binding homeobox 1 (*ZEB1*), *ZEB2*, and matrix metalloproteinase-1, which are indicative of EMT [72].

The notch pathway is extensively studied as a regulator in breast cancer EMT. Activation of the notch pathway induces the expression of the phosphatidylinositol 3-kinase/protein kinase B pathway, which play an important role in cell growth and proliferation by inducing the transcriptional activity of NF-κB [88]. NF-κB promotes EMT by affecting the downstream effectors, including MMP-9, VEGF, Cyclin D1, and Survivin [89,90]. The role of NF-κB in EMT is demonstrated by decreasing metastatic potential of epithelial cells in the presence of NF-κB inhibitor [91].

Cytokines are often released by the tumor cells and play important roles in regulating EMT. Irradiation of epithelial cells induces the production of interleukin-6, which activates the JAK/STAT pathway to induce the expression of various downstream effectors that promote EMT [92]. Tumor necrosis factor-α (TNF α) promotes metastasis and induces EMT in breast cancer cells by inducing the expression of Twist1 [93]. Transforming growth factor-β (TGF β) pathway is commonly associated in EMT in breast cancer. Hypoxia is commonly associated with EMT initiation in breast cancer. Hypoxia induces the expression of HES1 and HEY1 effectors via activation of Notch2. Upregulation of these effectors decreases the expression of E-cadherin, which is an important feature of EMT [94]. Hypoxia also affects the expression of cadherins to promote EMT via the expression of cyclooxygenase-2 [95].

MicroRNAs (MiRs) also play an important role in regulating EMT in breast cancer. For instance, downregulation of MiR-190 is frequently present in breast cancer tissues and shown to promote EMT and metastasis [96]. MiR-27a promotes EMT in breast cancer cells by affecting the expression of F-box and WD repeat domain containing 7 [97].

## 3. Nanotherapeutics

Novel drug delivery technologies encapsulating chemotherapeutics are extensively explored to target and treat cancer [98,99]. Nanocarriers, such as nanoparticles, liposomes, and micelles have been employed for cancer targeting, treatment, diagnosis, and imaging [100]. Chemotherapy mostly involves use of small molecular hydrophobic drugs that can be rapidly cleared from the tumor site due to their small molecular weight. This can result in lower circulation half-life and sub-therapeutic concentrations at the tumor site. Hydrophobic chemotherapeutic drugs can also pose a problem for their formulation since these require alcohol-based solvents for their dissolution. In addition, such chemotherapeutic agents are administered through the intravenous (i.v.) route and have difficulty with effective dissolution and release of the drug from the tumor site [101,102]. Gene therapies, such as siRNA, microRNA, or oligonucleotides for cancer treatment, suffer from systemic degradation when administered without a nanoparticle-based system. Nucleases present in the blood circulation can easily degrade naked siRNA and gene therapy products and significantly reduce their t ^1/2^ (<15 min) [103]. Nanocarrier-based drug delivery systems protect the chemotherapeutic agents from systemic degradation and allow targeted delivery to the tumor cells by active tumor-targeting. Nanocarriers not only protect chemotherapeutic drugs by its encapsulation but also protect the body from the toxic side effects of the chemotherapeutic drugs [104,105,106].

### 3.1. Passive Targeting

Tumors that grow more than 1–2 mm require blood vasculature for their nourishment supply. Such vasculature is often leaky and disorganized [107]. Tumor cells secrete vascular endothelial growth factor (VEGF) and other growth factors, such as matrix metalloproteinases (MMPs), bradykinin (BK), prostaglandins (PGs), and nitric oxide (NO), which promote surrounding endothelial cells for angiogenesis [108,109,110,111]. This leads to active accumulation of nanocarriers in the tumor site and subsequent release of payload due to leaky vasculature and lower lymphatic drainage. This phenomenon is called the enhanced permeability and retention (EPR) effect, first discovered by Matsumura and Maeda [112]. Exploiting the EPR-effect, nanocarriers can attain high drug concentrations at the tumor site and reduce toxic effects of the chemotherapy drugs on other organs. Nearly all types of nanocarriers, such as nanoparticles, liposomes, micelles, dendrimers encapsulating small molecules, proteins, peptides, and nucleotides, are capable of achieving the EPR effect [113,114]. Nanoparticles can accomplish sustained drug release at tumor tissue without overburdening the cells with large doses of chemotherapy. This passive targeting strategy can help to overcome MDR, commonly seen in resistant cancer cells [115,116]. Table 1 summarizes passive drug targeting-based nanomedicines currently undergoing evaluation [25]. Figure 4A illustrates passive drug delivery by the EPR effect [25].

### 3.2. Active Targeting

The EPR effect is effective in drug delivery and passive targeting of vascularized tumor tissue. Targeted drug delivery for avascular tumor cells, such as metastatic breast cancer, pancreatic, and prostate cancers, is challenging. Actively targeted nanocarrier systems utilize certain receptors that are overexpressed on cancer cells compared to normal cells for selective delivery of chemotherapeutic agents to the cancer cells. Actively-targeted nanocarriers can accommodate antibodies, peptides, polymers, DNA aptamers, and small molecules for selective detection and uptake into the cancer cells [126,127,128,129]. Among such targeting agents, antibody-targeting possesses high selectivity and specificity for breast cancer cells. Trastuzumab (anti-HER2 monoclonal antibody)-targeted magnetic polymersomes were able to target bone-metastasis in an HER2 positive mice breast cancer model [130]. NPs conjugated with trastuzumab were developed for active targeting, drug release, and imaging of metastatic breast cancer cells [131]. Monoclonal antibody conjugated superparamagnetic iron oxide NPs (SPIONs) were able to target the neu receptor in primary breast tumors in vivo [132]. Although, monoclonal antibodies can be a good tool for tumor-targeting, their large size can possess a formulation hurdle and alter NPs pharmacokinetic and pharmacodynamics properties. An equally effective alternative for antibody targeting can be the use of peptides. Arginylglycylaspartic acid (RGD) peptide [133,134,135], tumor metastasis targeting (TMT) peptide [136], P-selectin binding peptide [137], octreotide [138,139], and chlorotoxin [140] are some examples of peptides frequently used for targeting metastatic breast cancer. Integrin receptors present on breast cancer cells, specifically αvβ3 integrin receptor, mediates breast cancer metastasis. Cyclic RGD peptides target αvβ3 integrin receptors and, thus, can be useful for breast cancer tumor targeting. Chain-shaped SPIONs modified with RGD peptide resulted in superior targeting of αvβ3 integrin receptors due to the SPIONs’ geometrically enhanced multivalent docking. This resulted in reduction of lung and liver metastasis of breast cancer in vivo [134]. RGD peptide functionalized NPs can be a useful tool to delivery chemotherapy drugs, such as Doxorubicin (DOX) and Paclitaxel (PTX) [141,142]. Figure 4B illustrates cancer cells targeting achieved by active drug delivery mechanisms [25].

Hyaluronic acid (HA) is a naturally occurring ligand for cluster of differentiation-44 (CD44) receptors. CD44 is overexpressed on cancer cells, which aids in cell migration and invasion. HA functionalized NPs benefit in developing nanocarriers that demonstrate preferential tumor accumulation [143]. Zhang et al. demonstrated that HA-modified cationic nanoparticles might be promising in vivo for overcoming CYP1B1-mediated breast cancer MDR [144]. Wang et al. also demonstrated that HA grafted PEI-PLGA nanoparticles encapsulating gambogic acid and TRAIL plasmid were effective in treating TNBC in vivo [145]. Small molecules, such as bisphosphonates, AMD3100, and folic acid, established breast cancer tumor targeting. Bisphosphonates, such as alendronate, are used for the selective delivery of therapeutic nanocarriers to the bone microenvironment. Alendronate coated PLGA nanoparticle-encapsulated curcumin and bortezomib demonstrated higher localization, tumor reduction, and better imaging by these MPs in vivo in bone metastatic model of breast cancer [146]. C-X-C chemokine receptor type 4 (CXCR4) is overexpressed on metastatic breast cancer cells. AMD3100 is an antagonist to CXCR4 receptor and can actively target breast cancer cells. Zevon et al. demonstrated that short wave infrared (SWIR) emitting nanoprobes decorated with AMD3100 are able to preferentially accumulate in CXCR4 positive lung metastatic of breast cancer lesions when injected in an in vivo lung metastatic model of breast cancer [147]. Folic acid targets 4T1 breast cancer cells, which have the ability to metastasize to the lung, liver, bone, lymph nodes, and brain. Selenium NPs coated with folic acid showed potent antiproliferative effect against 4T1 cells in vivo [148]. Table 2 depicts the actively-targeted nanomedicines to overcome drug resistance and improve efficacy in breast cancer [25].

## 4. Nanomedicine to Overcome Drug Resistance in Breast Cancer

As discussed in the earlier sections, nanoparticulate drug delivery systems have been employed in the treatment of breast cancer for improving the efficacy of several chemotherapeutic agents. In addition to enhanced efficacy, nano-chemotherapeutics overcome drug resistance of the tumor and reduce the toxicity associated with the anti-cancer agents. Other advantages associated with nanoparticles include loading combination drugs in the same carrier, high drug loading, and controlled release. To date, several anti-cancer drugs have been encapsulated into nanocarriers, such as lipid nanoparticles, polymeric nanoparticles, lipid polymer hybrid nanoparticles, and micelles. Most widely encapsulated anti-cancer agents include DOX, PTX, and docetaxel. The majority of these carriers are still in pre-clinical stages; however, products such as Abraxane^®^ (paclitaxel loaded albumin nanoparticles; Bristol Meyers Squibb), Doxil^®^ (doxorubicin loaded in liposomes; Baxter), and BIND-014 (docetaxel loaded polymeric nanoparticles, Bind Therapeutics) are in clinical use, and several other nanocarriers are in various stages of clinical trials [160,161,162]. In addition to these, several other agents have also been encapsulated into nanocarriers for improved efficacy. The sections below summarize the progress on the use of the nanocarriers for the treatment of metastatic breast cancer.

### 4.1. Doxorubicin

Doxorubicin (DOX) is an anthracycline based anti-cancer antibiotic that acts by inhibiting topoisomerase II enzyme or reducing the oxidative stress in cancer cells. Clinical use of DOX is limited due to its dose-dependent cardiotoxicity. To date, DOX has been encapsulated into several nanocarriers that can be broadly classified as (i) inorganic nanoparticles (gold, silver, iron oxide, and others), (ii) organic nanoparticles (lipids and polymers), and (iii) integrated nanoparticles. Among these, inorganic nanoparticles have shown efficacy only in preclinical studies. Organic nanoparticles loaded with DOX include liposomes, polymeric nanoparticles, dendrimers, micelles, single-walled carbon nanotubes, and nano-diamonds with success in clinical studies [160,162]. Among the nanoparticles in development, liposomal DOX and pegylated liposomal DOX have been evaluated in phase II and phase III studies and marketed for clinical use. Encapsulation of DOX into liposomes can lead to alteration of tissue distribution and pharmacokinetics resulting in increased therapeutic index compared to conventional doxorubicin. Pegylated liposomes loaded with DOX can evade the mononuclear phagocyte system, resulting in prolonged half-life and increased circulation time of the liposomes. Apart from liposomes and pegylated liposomes, preclinical studies conducted with other nanocarriers have significantly enhanced the efficacy of DOX [163]. For example, in one study, nanodiamonds loaded with doxorubicin significantly reduced the drug efflux and tumor growth, along with increased apoptosis and inhibition of lung metastasis compared to conventional doxorubicin treatment in breast cancer [164,165].

Kaminskas et al. were successful in >95% reduction of lung metastasis with pegylated poly-lysine based dendrimers loaded with doxorubicin [166]. In another study, chlorotoxin-conjugated liposomes loaded with doxorubicin significantly inhibited the growth of 4T1 tumors in mice and prevented the lung metastasis [140]. In addition, doxorubicin polymeric micelles prepared using poly-L-lactide-PEG and poly-L-histidine-PEG and poly (acrylic acid)-g-PEG graft copolymers significantly reduced tumor growth and metastasis in mice with 4T1 tumors [167,168]. Polymer-lipid hybrid nanoparticle systems have also been proven to enhance the uptake and retention of doxorubicin in multidrug-resistant breast cancer cells. These studies claim that polymer-lipid hybrid nanoparticle systems successfully bypass the P-gp efflux transporters on the membrane of cancer cells, resulting in protection against drug resistance [169]. Sun et al. prepared DOX loaded low molecular weight heparin (LMWH) based LMWH-Cholesterol (LHC) conjugates for intravenous delivery. The nanoparticulate system had a longer circulation time compared to doxorubicin alone, and in vitro results confirmed that the nanoparticles were more effectively taken up by 4T1 cells and showed a stronger anti-metastatic effect by cell invasion and cell migration compared with doxorubicin alone [170]. It is also reported in the literature that nanoparticulate based drug delivery systems enhance the efficacy of combination drugs. Shuhendler et al. co-encapsulated doxorubicin and mitomycin C into a polymeric lipid hybrid nanoparticle system and achieved synergistic effects with the combination compared to single drugs. Results showed that nanoparticles killed the cancer cells at 20–30 times lower doses compared to single drugs [171]. In another study, a polymer-lipid hybrid nanoparticle system co-encapsulating doxorubicin and GG918 (Elacridar) significantly enhanced the uptake of the drugs compared to single agents [172].

### 4.2. Paclitaxel 

Paclitaxel (PTX) is a lipophilic anti-cancer drug isolated from the Pacific yew tree (Taxus brevifolia). PTX is a hydrophobic anti-cancer compound, widely used as a first line agent for the treatment of breast cancer. To enhance the solubility of PTX, it is formulated as Taxol^®^ (Bristol Meyers Squibb, New York, NY, USA), containing Cremophor EL and dehydrated ethanol (50:50, *v*/*v*) as solubilizing agents. However, this formulation is associated with side effects such as hypersensitivity, neuropathy, and neurotoxicity. Therefore, a nanocomplex (Abraxane^®^) was developed using serum albumin and marketed as Abraxane^®^ [173]. Abraxane^®^ is a solvent free formulation containing an albumin-bound form of paclitaxel. Advantages include delivery of paclitaxel at higher doses over a shorter infusion time, enhanced endothelial transport, and prevention of hypersensitivity reactions [174,175,176]. Several other formulations, such as liposomes, polymeric nanoparticles, polymeric micelles, nanocrystals, and others, have been developed to enhance the efficacy of PTX [177]. Tang et al. developed vitamin E (VE)-albumin core-shell nanoparticles loaded with PTX for enhancing the efficacy in multidrug-resistant breast cancer. The nanoparticulate formulation evaluated in the study showed stronger cytotoxicity and increased accumulation in breast cancer cells. In addition, an in vivo study in a xenograft model showed higher anti-cancer activity with nanoparticles compared to solution formulation [173]. In another study, PTX-loaded folate-conjugated cyclodextrin nanoparticles enhanced the uptake of PTX via receptor-mediated endocytosis. The folate-conjugated nanoparticles were better tolerated in the breast cancer model with significant reduction in tumor burden and can be considered a promising alternative to other PTX formulations [178]. Pegylated PTX nanocrystals prepared using the antisolvent precipitation technique showed higher stability and significantly better antitumor activity (82% tumor reduction; *p* < 0.05) compared to PTX nanocrystals [177]. Xu et al. significantly enhanced the anti-cancer activity of PTX by incorporating it into solid lipid nanoparticles in MCF7 and MCF7/ADR cell lines. Encapsulation of PTX into solid lipid nanoparticles successfully reversed the multidrug resistance by evading the efflux pumps in drug-resistant cells [179]. In addition to these formulations, co-encapsulation of PTX with other drugs into nano-formulations significantly enhanced the anti-cancer activity of the compounds in metastatic/multidrug-resistant breast cancer cells. The co-encapsulated drugs include curcumin, lonidamine ceramide, and antagomir-10b [180,181,182,183].

### 4.3. Docetaxel

Docetaxel is a lipophilic anti-cancer agent derived from a European yew tree (Taxus baccata). It is a cytostatic agent that acts by reversibly binding to microtubules, promoting transitory structure stabilization and, ultimately, causing cell cycle arrest [184]. Oliveira da Rocha et al. prepared solid lipid nanoparticles loaded with docetaxel using Compritol 888 ATO as the solid lipid. Encapsulation of docetaxel into solid lipid nanoparticles resulted in 100× reduction of IC_50_ and enhanced uptake into the cells compared to docetaxel alone. In addition, in vivo efficacy studies showed that docetaxel loaded solid lipid nanoparticles exhibited higher anti-cancer activity with significant reduction in tumor volume (*p* < 0.0001) and prevention of lung metastasis [184]. In another study, polymeric micelles loaded with docetaxel were prepared using poly (D, L-lactide)1300-b-(polyethylene glycol-methoxy)2000 (mPEG_2000_-b-PDLLA_1300_). Polymeric micelles showed similar efficacy in terms of growth suppression of primary tumors but greater chemotherapeutic efficacy against breast cancer metastasis compared to docetaxel alone [185]. Similar tumor reduction was observed in another study, where docetaxel was encapsulated into polymeric micelles (NC-6301) prepared using poly (ethylene glycol)-poly(aspartate) block copolymer [186]. Xu et al. developed a shrapnel based liposomal system with reduction and enzyme sensitive properties loaded with docetaxel. Liposomes were prepared using methoxy polyethylene glycol-peptide-vitamin E succinate and were sensitive to matrix metalloproteinases in the tumor microenvironment for the release of docetaxel. Compared to docetaxel alone, increased distribution of docetaxel was observed in lungs and tumors of 4T1 tumor-bearing mice. In addition, tumor growth and pulmonary metastasis were inhibited due to enhanced docetaxel induced apoptosis and the reduced metastasis-promoting protein expression [187]. Similar reduction of tumor growth, enhanced uptake, and prevention of metastasis with docetaxel was achieved with other nano-formulations, such as nanoliposomes [188], albumin conjugates [189], Ecoflex^®^ nanoparticles [190], and human serum albumin nanoparticles [191]. In addition to these formulations, co-encapsulation of docetaxel with other drugs into nano-formulations significantly enhanced the anti-cancer activity of the compounds in metastatic/multi drug-resistant breast cancer cells. The co-encapsulated drugs include quercetin [192], IGF-1R siRNA [193], cisplatin [194], miRNA-34a [195], cMET siRNA [196], and thymoquinone [197].

### 4.4. Other Drugs

In addition to PTX, docetaxel, and DOX, nanoparticulate systems have also been reported for the delivery of drugs such as adriamycin [198], saporin [199], wedelolactone [200], curcumin [201], irinotecan [202], siRNA [203], succinobucol [204], cisplatin [205], probucol [206], artemisinin [207], and silibinin [208]. These studies suggest that loading these drugs into nanoparticulate drug delivery systems has significantly enhanced their therapeutic potential in metastatic breast cancer. This improvement provides great opportunity for easily encapsulating several other anti-cancer drugs approved by the USFDA into nanoparticles. Such nanocarriers can deliver drugs at higher concentrations specifically to the tumor site. Tumor targeting delivery may reduce toxicity to healthy cells. However, the binding domain of efflux transporters present inside the cell membrane. Therefore, once the drug is released from the nanocarriers in the cytoplasm, it can be refluxed out unless an inhibitor is present. If nanocarrier-mediated delivery surpasses the transporters’ efflux capacity, it may achieve some fruitful results. However, it cannot overcome drug resistance.

### 4.5. Combination Chemotherapy 

The rationale for combining chemotherapeutic small molecular drugs is centered on targeting various biochemical pathways to overcome MDR, specifically in heterogeneous tumors such as breast cancer [209]. Targeting a specific mechanism with a single chemotherapy drug can lead to activation of alternative metabolic pathways. This frequently contributes to emergence of MDR [210]. This therapy is based on combining and delivering chemotherapy drugs that work by non-overlapping molecular mechanisms, thus, reducing MDR occurrence and maximizing tumor killing [211]. This concept of using combination chemotherapy to improve the clinical efficacy with reduced clinical toxicity has greatly evolved in the last decade. Preclinical studies demonstrate that chemotherapy is most effective when administered in combinations aiming to achieve additive or synergistic effect and reduce MDR [115,116,211].

Combining various chemotherapy drugs in multifunctional nanocarriers in a predetermined ratio can achieve effective and predictive delivery of the drugs at the tumor site. Coupled with tumor active targeting, enhanced uptake and lower off-target effects are evident [212,213,214]. Tang et al. demonstrated that co-delivery of epirubicin and paclitaxel, encapsulated in estrone-targeted PEGylated nanocarriers, significantly suppressed tumor growth in vivo in MCF-derived mouse model with minimal toxicity [215]. Paclitaxel and cisplatin encapsulated in poly (2-oxazoline) polymeric micelles showed improved drug release, drug loading, pharmacokinetics, and efficacy in MDR breast cancer LCC-6-MDR orthotopic tumor model and MDR human ovarian carcinoma xenograft tumor [216]. pH-sensitive micelles encapsulating docetaxel and silibinin displayed enhanced higher toxicity, cellular uptake, and stronger anti-metastasis effect in mouse breast cancer cell line 4T1. 4T1 tumor-bearing mice, when treated with the micelles, demonstrated inhibition on breast cancer growth and metastasis [217]. Dong et al. showed that co-delivery of paclitaxel and gemcitabine by peptide targeted PLGA NPs had improved anti-cancer effect and reduction in systemic toxicity compared to the free drugs in vivo [218]. 

Co-delivery of Docetaxel and Thymoquinone entrapped in chitosan targeted lipid nanocapsules resulted in enhanced chemotherapeutic efficacy in MCF-7, MDA-MB-231 cells, and resistant human breast cancer cells [219]. Lu et al. demonstrated that co-delivery of Cyclopamine and Doxorubicin in albumin NPs could target primary breast tumors, target the metastatic lymph nodes, and simultaneously inhibit the tumor metastasis in vivo. This study also showed that the NPs could reverse Doxorubicin resistance in breast cancer chemotherapy [220]. Lan et al. engineered paclitaxel and capsaicin prodrug micelle for breast cancer drug delivery. The co-drug loaded micelles demonstrated superior in vivo antitumor activity in mice and reduced the tumor growth rate [221]. Co-delivery of berberine and doxorubicin using PLGA NPs demonstrated enhanced in vivo anti-cancer activity and kinetics in Sprague-Dawley rats [222]. Wang et al. demonstrated that co-delivery of paclitaxel and honokiol in pH-sensitive polymeric micelles can suppress MDR in TNBC. The micelles demonstrated P-gp inhibition, pH-triggered drug release, and MMPs inhibition in vivo in mice model of TNBC [223]. In combination therapy, one agent can act as a competitive inhibitor for efflux transporter allowing a second anti-cancer agent to be retained its anti-cancer effect. For example, in a paclitaxel and cisplatin combination where cisplatin can inhibit efflux of paclitaxel or vice versa. Similarly, when epirubicin and paclitaxel are combined, epirubicin can act as an efflux inhibitor, increasing the intracellular levels of paclitaxel. Such combination targeted therapy can produce therapeutic efficacy by overcoming drug resistance.

### 4.6. Nanotherapeutics for Triple Negative Breast Cancer

TNBC is characterized by low expression of ER, PR, and HER2 receptors [224]. TNBC is unresponsive to conventional chemotherapy due to the lack of molecular targets and aggressive phenotypes. Hence, NPs based chemotherapeutics can be an effective alternative for targeting certain receptors on TNBC and providing effective drug delivery to the cancer cells [225]. Multifunctional NPs having abilities such as (i) targeted drug delivery and (ii) noninvasive imaging for tumor cells, and uptake of nanotherapeutics, such as theranostics NPs, hold great potential toward the development of novel TNBC nanotherapeutics [225]. Certain receptors, such as CD44, are overexpressed on TNBC cells. HA, being a ligand for these receptors, can be suitable for active targeting of NPs to treat TNBC [226,227,228,229]. CXCR4 is another receptor overexpressed on TNBC cells. Similar to CD44, the CXCR4 receptor is also involved in the growth, metastasis, and progression of TNBC. Plerixafor (or AMD3100) is a small molecule ligand for the CXCR4 receptor. Plerixafor targeted NPs demonstrated improved cellular accumulation and uptake in MDA-MB-231 cells and improved siRNA-mediated gene silencing in vivo [230]. Urokinase plasminogen activator receptor (uPAR) was also explored as a target for TNBC. Peptide targeting uPAR decorated NPs constructed from PLGA polymer were developed for delivering two antisense miRNA (antimir-10b and antimiR-21). These NPs showed higher TNBC tumor inhibition in vivo than the scrambled peptide decorated NPs [231].

Many nanotherapeutics, such as liposomes, micelles, solid lipid NPs, polymeric NPs, and dendrimers were investigated for improving bioavailability and reducing clearance in TNBC. In recent years, combinatorial nanotherapeutics have shown advantage over traditional single treatments for TNBC treatment in preclinical studies. In this therapy, different molecules, small molecules, or gene therapy were loaded into the NPs [231,232]. Layer-by-layer NPs loaded with siRNA and DOX were studied by Deng et al. The scientists observed a silencing of MRP-1 by the siRNA. This caused an increase in efficacy of DOX four-fold in vivo. This caused a rapid decrease in tumor volume by eight times with minimal to no toxic effects [232]. Combinatorial drug therapy can also involve combination of small molecule chemotherapeutics with photothermal therapy. Such NPs systems are called “theranostics”. Such systems can simultaneously treat, diagnose, and image cancer tumors in one single NP system. Theranostics is a multipurpose system in which a localized NIR laser illumination can generate heat to inhibit TNBC tumors and allow release of cisplatin from cisplatin loaded gold nanorods. These nanorods with a photothermal and chemotherapy combination demonstrated suppression of TNBC metastases to the lungs in vivo [233]. A layer-by-layer NP was engineered for systemic co-delivery of DOX and siRNA. SiRNA was deposited by alternating films of poly-L-arginine. It was observed that bilayers on NPs surface were able to load up to 3500 siRNA molecules. This greatly enhanced the serum t^1/2^ of the siRNA [232]. Meng et al. demonstrated that co-delivery of DOX and P-gp inhibiting siRNA encapsulated in mesoporous silica NPs allowed them to overcome drug resistance in vitro in MCF-7/MDR cells and in vivo in MCF-7/MDR xenograft model in nude mice [234]. This study is the first detailed analysis of breast cancer heterogeneity in the tumor microenvironment and the efficacy of siRNA delivery systems in vivo [234]. Parvani et al. developed a lipid ECO-based NP delivering β3 integrin siRNA (ECO/siβ3). The β3 integrin is linked to epithelial-mesenchymal transition (EMT) and metastasis in GTNBC and many other cancers as well. ECO/siβ3 NPs were modified by targeting with an RGD peptide via a PEG spacer for enhanced siRNA uptake and accumulation in TNBC cells. The RGD-targeted ECO/siβ3 NPs alleviated primary TNBC tumor burden and significantly inhibited metastasis in vivo in MDA-MB-231 TNBC induced mice [235]. Su et al. constructed a “triple punch” NP system for TNBC treatment with a combination of three drugs, each serving a different purpose. Indocyanine green (ICG), paclitaxel (PTX), and survivin siRNA were encapsulated into a thermosensitive NP system. Controlled release of PTX in TNBC tumor was triggered by laser irradiated ICG, which produced local hyperthermia. Survivin siRNA can show remarkable inhibition of tumor metastasis. When combined with chemotherapy, it can enhance the sensitivity of TNBC tumor cells to chemotherapy [236]. This “triple punch” NP system exhibited remarkable antitumor activity due to the combinatory effects of photothermal therapy, chemotherapy, and gene therapy having low drug doses, which were effective in reducing MDR [237]. Figure 5 discusses the results of in vivo antitumor activity of “triple punch” NPs in a xenograft model of human breast cancer in BALB/c athymic nude mice [237].

### 4.7. Stimuli Responsive Drug Release 

Stimuli-responsive drug delivery systems (DDS) can release their payload on exposure to external stimuli, such as physical, chemical, or biological responses. Stimuli responsive or “Smart” DDS can achieve target-activated release of the drug in the vicinity of the tumor or tumor cells. This enables reduction of systemic toxicity by chemotherapeutic agents and limiting exposure of healthy tissues to the cytotoxic drugs. Exploitation of physiological differences between tumor tissue and healthy tissue aids in the design of stimuli-responsive DDS. Stimuli-responsive DDS can further be classified as internal stimuli assisted and external stimuli assisted drug release [25].

#### 4.7.1. Internal Stimuli Assisted Drug Release

Inherent differences between the normal tissue and tumor microenvironment are present, which can be explored and utilized for engineering an internal stimulus assisted DDS.

##### pH-Responsive Drug Release

The pH of the tumor microenvironment is remarkably different from the surrounding healthy tissues. pH conditions in the tumor site are more acidic (pH 6.5–7.2) than in the healthy neighboring tissue (pH ~ 7.4). In addition, pH of the endosomes and lysosomes is typically even more acidic (pH 4.5–6.8) than the tumor microenvironment. These differences in the pH can be harnessed for pH-responsive drug release from the DDS in tumor tissue [238,239,240,241]. Several studies have shown the benefit of this strategy at the preclinical level for effective delivery and drug release of the stimuli-sensitive DDS in the tumor tissue. Jia el.al. established pH responsive multifunctional mesoporous silica NPs with co-delivering TET, an MDR reversal agent, and PTX for breast cancer treatment. These NPs demonstrated a significant inhibition of breast cancer cell proliferation and a P-gp-dependent MDR reversal in MCF-7/ADR cells [240]. Li et.al developed a polymersome constructed from amphiphilic polypeptide-based pH-sensitive block copolymer encapsulating DOX and verapamil. This DDS showed reversal of MDR in MCF-7/ADR breast cancer cells [242]. Liu et al. developed a pH-sensitive, dual functionalized acid micelle, which could deliver PTX to breast cancer tumors and also result in reversal of MDR. The micelles were targeted with HA. HA is the ligand for CD44 receptors, overexpressed on metastatic breast cancer cells. The hyaluronic acid-deoxycholic acid (HA-DOCA)-His micellar delivery system used in this study had dual properties. HA provided active targeting while DOCAS lipid polymer caused endosome pH-triggered drug release. Cytotoxicity of PTX in HA-DOCA-His micelles in drug-resistant breast cancer (MCF-7/MDR) was improved significantly. In addition, MDR-overcoming in MCF-7/MDR cells was observed compared to Taxol treatment. Interestingly, PTX loaded in the HA-DOCA-His micellar system was more effective in breast cancer inhibition in MCF-7/Adr tumor-bearing mice [243]. In addition, Figure 6 portrays the in vivo antitumor activity of Taxol, PTX/HA-DOCA, and PTX/HA-DOCA-His nanomicelles in a MDR human breast carcinoma MCF-7/Adr tumor model [193].

A defective PI3K/AKT/mTOR signaling pathway is known to cause MDR and cancer metastasis. Yin et al. designed a pH-sensitive nanocomplex co-delivering PTX and a siRNA metastatic breast cancer. The siRNA could silence the Akt expression in metastatic breast cancer 4T1 cells. PTX-loaded micelle/siAkt nanocomplex (PMA) was able to downregulate P-gp, upregulate Caspase-3 expression, cause Akt gene downregulation, and knockdown in 4T1 cells. In addition to the excellent in vitro results, PMA also demonstrated efficacy and safety in vivo. In 4T1 tumor-bearing mice, PMA achieved a 94.1% tumor inhibition and suppressed 96.8% lung metastases of breast cancer. The nanocomplex PMA had very low toxicity and did not cause lesions in normal organs [244] (Figure 7). Cheng et al. developed a pH sensitive micelle from a pH-sensitive polymer (POT) encapsulating DOX. The micellar system contained α-tocopheryl succinate, which is known to generate reactive oxygen species (ROS) in cancer cells. DOX-loaded POT micelles possessed highest drug accumulation and the strongest tumor growth inhibition in breast cancer induced mice [245]. In addition, the micelles could induce higher percentage of apoptosis at the tumor site without damage to healthy tissues and reduced breast cancer metastasis in vivo [245]. Cheng et al. observed similar results in NPs co-delivering DOX and pyrrolidinedithiocarbamate (PDTC). PDTC acts as a chemosensitizer and can efficiently silence P-gp expression while increasing intracellular drug levels by inhibiting the NF-κB pathway [246,247]. The researchers designed pH-sensitive NPs that were based on poly (ortho ester urethanes) copolymers. These copolymers had ortho-ester bonds that are stable in a neutral pH environment but rapidly degrade under mildly acidic conditions [248,249]. Results from this study, such as monolayer and multicellular spheroid (3D) experiments, demonstrated that PDTC was able to reverse MDR, enhance intracellular DOX accumulation, and downregulate P-gp expression. This resulted in higher DOX-induced apoptosis in MCF-7 and MCF-7/ADR cell lines. Higher DOX accumulation and greater tumor growth inhibition up to 83% was seen in MCF-7/ADR bearing-mice treated with the NPs [250].

#### 4.7.2. Redox-Responsive Drug Release

The redox potential of glutathione (GSH) is often used for intracellular stimuli responsive drug release from DDS or prodrugs. Intracellular concentration of GSH in cancer cells is ~10 mM, which is very high compared to its concentration (~2 μM) in the extracellular milieu [251]. Thus, this difference between tumor and normal cells can be harnessed by DDSs for effective delivery and release of cytotoxic agents in the tumor microenvironment. Disulfide bonds can be reduced to thiol bonds by GSH present in tumor cells [252]. Thus, a DDS with a disulfide bond in their structure can accelerate the release of the chemotherapeutic agent in the tumor tissue by activation by intracellular GSH. This also results in reduced toxicity to other healthy tissues due to insufficient GSH mediated drug release [253,254,255,256]. A recent study demonstrated that redox and pH-sensitive podophyllotoxin (PPT) prodrug micelles could be used for reversing MDR in breast cancer. The micelles were designed to target transferrin receptors. The drug was covalently bonded to T7-peptide (Pep) through a disulfide bond. In vivo results of the micelles showed enhanced antitumor activity against MCF-7/ADR xenograft mice compared to the control group [257]. Rajendrakumar et al. developed a theranostic nanoassembly, having combinatorial therapies coupled with real-time monitoring, for breast cancer treatment and diagnosis. The HA-stabilized redox-sensitive polyplex (HART) encapsulated DOX intercalated Bcl-2 shRNA. HART nanoassembly could achieve CD44-mediated intracellular uptake in MCF7 breast cancer cells and redox-responsive drug-gene release. The HART nanoassembly also contained a dual MRI contrast (T1/T2) agent and demonstrated efficacy in vitro [258]. Qiao et al. designed a redox-triggered micelle encapsulating mitoxantrone prodrug for overcoming MDR in breast cancer. An in vitro cytotoxicity study on MDA-MB-231/MDR cells demonstrated inhibition of growth and development of resistance in TNBC. In addition, the redox-sensitive micelles showed stronger antitumor activity in xenograft mice with minimal side effects [259]. Li et al. designed a micelle system from polymer containing a disulfide bond, encapsulating PTX and dasatinib. The co-loaded micelles demonstrated good cytotoxicity and sensitivity towards MCF-7/ADR cells [260]. Figure 8A illustrates mechanisms for internal stimuli assisted drug release [25]. We have also developed hyaluronic acid (HA) decorated mixed nanomicelles encapsulating paclitaxel (PTX) and P-gp inhibitor ritonavir (RTV). HA was conjugated to poly (lactide) co-(glycolide) (PLGA) polymer by disulfide bonds, (HA-ss-PLGA). Addition of RTV inhibits P-gp and CYP3A4 mediated metabolism of PTX, preventing MDR and sensitizing the cells towards PTX. In vitro uptake and cytotoxicity study in MBC, MCF-7, and TNBC MDA-MB-231 cell lines demonstrated effective uptake of the nanomicelles and drug PTX compared to MCF-12A (normal breast) cells, while cell viability assay indicated 75% survival in MCF-12A cells compared to 25% in MCF-7 cells after treated with HA-PTX + RTV NMF for 72 h. An in vitro potency determination indicated reduction in mitochondrial membrane potential and decrease in reactive oxygen species in breast cancer cell lines, indicating effective killing of the cancer cells while sparing healthy cells. Therefore, stimuli sensitive and redox responsive nanomicelles, along with HA targeting and RTV, can effectively serve as a chemotherapeutic drug delivery to overcome MDR in breast cancer [261].

#### 4.7.3. External Stimuli-Responsive DDS

External stimuli-responsive DDS includes release of payload by physical triggers, such as temperature [262,263], electromagnetic stimuli such as photodynamic therapy [264], ultrasound [265], electric field [266], and applied mechanical force [267]. Jose et al. developed temperature-sensitive liposomes encapsulating tamoxifen and imatinib drugs for breast cancer treatment. The temperature-sensitive liposomes demonstrated more than 80% drug release within 30 min in MCF-7 and MDA-MB-231 breast cancer cells when temperature was increased to 39.4 °C [268]. A temperature-sensitive phase-change hydrogel containing tamoxifen (Tam-Gel) indicated effective local release of the hormone agonist in ERα-positive breast cancer. The hydrogel demonstrated sol-gel transformation at room temperature. The hydrogel decreased the intra-hepatic growth of breast cancer metastasis and reduced tumor growth in vivo [269]. Biocompatible piezoelectric NPs were capable of targeting and stimulating HER2-positive breast cancer cells. These NPs were activated by externally delivered ultrasound stimulus. These externally stimulated NPs significantly reduced proliferation in vitro by inducing cell cycle arrest. These NPs were able to lower breast cancer proliferation by upregulating inward rectifier-potassium channels by meddling on Ca^2+^ homeostasis. Additionally, these NPs also were effective by increasing expression of the gene encoding for Kir3.2 [270]. A couple of studies reported use of ultrasound-responsive DDSs delivering PTX for MDR in breast cancer. The first study demonstrated the development of ultrasound-sensitive nano liposomes encapsulating PTX and Bcl-2 siRNA. These liposomes effectively partitioned into the vasculature and tumor tissue by external low-frequency assisted ultrasound stimuli and allowed enhanced intracellular accumulation of PTX and Bcl-2 siRNA [271]. The second study reported the development of multifunctional microbubbles laden with oxygen and PTX. These microbubbles were highly effective in reversing MDR and reducing tumor size in vivo in human ovarian cancer xenograft mice model [272]. Oxygen supplied through these microbubbles could decreased expression of HIF-1α and of P-gp due to increased tumor oxygenation [272]. Figure 8B illustrates mechanisms of externally activated drug release [25].

### 4.8. Breast Cancer Stem Cell Targeting Nanotherapeutics

Cancer stem cells (CSCs) are a subpopulation of cells found in tumor tissue with capabilities of self-renewal, differentiation, and tumorigenicity [273]. CSCs also play a key role in tumor metastasis and MDR. Targeting this population of cells in a tumor can be explored as a therapeutic strategy to reduce tumor progression, metastasis, tumor relapsing, and MDR. However, targeting CSCs is likely challenging due to the heterogeneous nature of the cancer and difficulty in targeting and selective inhibition of CSCs by currently available therapeutics. CSCs can be characterized from other cancer cells by expression of surface markers such as CD44, CD133, and CD24 [274]. CD44 binds to HA present in the extracellular matrix (ECM) and aids in attachment of CSCs to the ECM. This causes proliferation and migration of CSC [275]. Breast cancer stem cells (BCSCs), similar to CSCs, are relatively resistant to conventional cancer therapies targeting the tumor bulk. CD44^+^/CD24^−^ phenotype is considered a characteristic of BCSCs. Innovative approaches targeting BCSCs and the various pathways regulating BCSCs have demonstrated promising results in preclinical settings [276,277,278].

Hormone positive breast cancers expressing ER and/or PR are further divided into Luminal A and Luminal B types. In these types, expansion and migration of BCSCs are due to PR-induced receptor activator of the NF-κB ligand (RANKL) and ER-induced paracrine FGF/FGFR/Tbx3 signaling pathway [279,280]. This suggests that breast cancer therapies need to be developed not only targeting breast cancer cells but also BCSCs. Karthik et al. demonstrated that mTOR inhibitors, such as rapamycin, everolimus, and PF-04691502 (a dual PI3K/mTOR inhibitor), in combination with tamoxifen, showed significant improvement and tumor shrinkage of ER-positive breast cancer [281]. The mTOR of such can result in shrinkage of the cells [281]. The mTOR pathway and cyclin D-CDK4/6 complex play significant roles in the regulation of BCSCs activity [281,282]. Gao et al. incorporated docetaxel and salinomycin in NPs for targeting breast cancer cells and BCSCs. Such NPs could maintain the synergistic ratio of the drugs, demonstrating higher tumor targeting and antitumor activity in vivo [84].

HER2 positive breast cancer has an aggressive biologic behavior and frequently results in metastasis. HER2 targeting agents, such as trastuzumab, lapatinib, pertuzumab, and trastuzumab emtansine (T-DM1), have greatly improved clinical outcomes for patients with HER2-positive breast cancer. Trastuzumab was not effective in targeting BCSCs that had low surface HER2 receptors. Hence, they utilized trastuzumab emtansine (T-DM1), an antibody drug conjugate, along with trastuzumab on CD44^+^/CD24^−^/HER2^−^ BCSCs. The study demonstrated that BCSCs were sensitive to T-DM1 [283]. Although, HER2-targeting agents like T-DM1 display high efficacy initially, most patients eventually develop resistance [284,285]. Li et al. showed that salinomycin lipid hybrid anti-HER2 NPs could target HER2-positive breast CSCs and cancer cells. These NPs reduced the breast tumor formation rate and BCSCs more effectively in vivo than non-targeted nanoparticles or salinomycin alone [286]. Salinomycin NPs decorated with EGFR and CD133 aptamers also effectively targeted osteosarcoma cells and BCSCs. It also inhibited tumor growth more than other controls in osteosarcoma-bearing mice [287]. Targeting CXCR1/2 receptors reduced BCSC activity in HER2 positive breast cancer ex vivo in metastatic and invasive human breast cancers [288]. TNBC has the highest percentage of BCSCs compared to the other breast cancer subtypes.

TNBC has no hormone receptors or HER2 receptors, hence, developing targeted therapies for such cancer is challenging. Chemotherapeutic drugs are the only ones used for TNBC. Although patients show positive cancer reduction initially, they later develop resistance and relapse occurs. This is the reason for the very low 5-year survival rate of TNBC. BCSCs having CD44^+^/CD24^−^ tumor-initiating and self-renewing capacities are primarily responsible for TNBC resistance and relapse [289]. Hence, novel innovative therapies that target TNBC stem cells, along with the susceptible cells, are badly needed. BCSCs demonstrate interconvertible epithelial-like or mesenchymal-like states. HA is a ligand for CD44 receptors overexpressed on BCSCs. HA-conjugated pH-sensitive NPs encapsulating curcumin and PTX drugs may enhance therapeutic efficacy to MDA-MB-231 stem cells and could lower progression of TNBC in vivo [290]. Cyclophosphamide, an inhibitor of the hedgehog-signaling pathway of CSCs and doxorubicin, was loaded into HA-PLGA NPs, and developed for targeted therapy for CSC. These NPs diminished the number and tumor size in vivo and reduced BCSCs [291]. Sulaiman et al. verified that dual inhibition of Wnt and Yes-associated protein signaling in BCSCs can effectively reduce the population of BCSCs in a human xenograft model [292]. They discovered that TNBC patients’ samples expressed higher levels of HDAC mTORC1 genes compared to samples with luminal breast cancer. In addition, co-inhibition of HDAC mTORC1 with valproic acid and rapamycin, respectively, promoted ESR1 expression in TNBC cells. A cocktail of drugs, such as valproic acid, rapamycin, and tamoxifen (ESR1 inhibitor), was significantly decreased in human TNBC xenograft model with BCSCs population [293]. Table 3 discusses various BCSCs targeting therapies in the clinical trials [294]. Figure 9 illustrates BCSCs and their roles and therapeutic implications [294].

TNBC, triple-negative breast cancer; ER, estrogen receptor; HER2, human epidermal growth factor 2; AI, aromatase inhibitor; DLT, dose-limiting toxicity; MTD, maximum tolerated dose; AE, adverse events; ORR, overall response rate; PFS, progression-free survival; mPFS, median PFS; OR, objective response; pCR, pathologic complete response; cCR, clinical complete response; OS, overall survival; NA, not applicable.

## 5. Recent Advancement in Breast Cancer Treatment

### 5.1. Targeting miRNA

MicroRNA (miRNA) are small RNAs that are about 22 nucleotides in length and bind to the non-coding region of the target mRNA. They promote degradation or inhibit translation and, thereby, decrease the target gene expression. miRNAs play important roles in tumor growth, metastasis, and cancer progression. Downregulation of miR-140 decreased the inhibition on Wnt/β-catenin pathway, which increased mammosphere formation and promoted breast cancer progression [295]. Dysregulation of miR-29b-1-5p was involved in the development of breast cancer. Breast cancer cells exposed to chemotherapeutic agents released extracellular vesicles loaded with various miRNAs (miR-9-5p, miR-195-5p, and miR-203a-3p) that promoted cancer stem cell like phenotype in treatment sensitive breast cancer cells [296]. These studies indicate a huge potential of miRNAs in breast cancer therapy.

Ectopic expression of miR-29-1-5p in triple negative breast cancer cell lines decreased the number of mammospheres, migration, and invasiveness by affecting the Wnt/β-catenin signaling pathway. Furthermore, sensitivity to paclitaxel treatment was increased in these cell lines [297]. Dysregulation of other miRNAs, including let-7, miR-600, miR-146, etc., increased tumorigenic potential by affecting the Wnt/β-catenin pathway. The miR-34a targeted the signaling molecules of Notch1 pathway to increase the sensitivity to PTX [298]. Transfection of various breast cancer cell lines with miR302/367 cluster decreased the population of cells in the S phase. It also affected the expression of various signaling molecules in the canonical and non-canonical pathways that regulate TGF-β gene expression [299]. Expression of miR-127 is downregulated in breast cancer tissues and is correlated with the patient survival. Umeh-Garcia et al. have developed a novel, bioengineered miR-127 prodrug (miR-127^PD^) that is processed to a mature form inside the cancer cells. Application of miR-127^PD^ decreased the viability of many triple negative breast cancer cells and increased the sensitivity to anti-cancer drugs. Further, administration of miR-127^PD^ intravenously into orthotopic xenograft NOD/SCID mice resulted in a decreased tumor volume and metastasis [300]. ALDH1 is expressed on BCSCs and is directly correlated with metastasis. Transduction of CD44^+^/CD24^−/low^ BCSCs with lenti-miR-7 has decreased the expression of ALDH1A3, CD44, and epithelial cell adhesion marker. Further, miR-7 transduction shrunk the tumors over time, indicating the potential for the development of new therapies that are directed to affect ALDH1 expression [301]. Therefore, miR-1 plays a vital role in mitochondrial respiration, and it is downregulated in BCSCs. Overexpression of miR-1 resulted in the mitochondrial damage of BCSCs by repressing the mitochondrial inner membrane organizing system and glycerol-3-phosphate dehydrogenase genes [302]. Overexpression of miR-489 increased the sensitivity of BSCSs to 5-florouracil treatment by targeting a key anti-apoptotic protein (X-linked inhibitor of apoptosis protein) [74]. Combination of trastuzumab with miR-200 overexpression decreased the number of mammospheres with CD44^+^/CD24^−/low^ BCSCs population in HER2 negative and positive breast cancer cells [303]. 

Exosomes are extracellular vesicles released from the cells that form an important reservoir of various miRNAs, which are responsible for affecting the disease progression. For instance, exosomes isolated from treatment resistant breast cancer cells induced epithelial-mesenchymal transition (EMT) [77]. EMT is a process that converts polarized in treatment sensitive cells through the transfer of miR-155 [304]. Adipose mesenchymal stem cell-derived miR-1236 in exosomes sensitized the breast cancer cells to cisplatin treatment by inhibiting the Wnt/β-catenin signaling [305]. Exosomes obtained from human umbilical cord mesenchymal stem cells exhibit inhibitory effects on the breast cancer cells in vitro and in vivo. The effect is mediated by the transfer of miR-148b-3p from exosomes to the breast cancer cells where it inhibits the gene expression of tripartite motif 59, which plays an important role in the development of cancers [306]. 

Novel formulation approaches are explored to use miRNAs in the treatment of breast cancer. For instance, chitosan formulation of docetaxel and anti-miR-21 had significant effect on breast cancer cells in vitro compared to either treatment alone [307]. Herceptin-conjugated cationic liposomal formulation of let-7-miR and CDK4 specific siRNA decreased the cell viability and decreased the migration of SK-BR-3 cells [308].

### 5.2. Precision Medicine in Breast Cancer

Precision medicine utilizes technology to identify the subpopulations within a disease group. This information is used to formulate treatment strategies that are specific to the individual. The United States introduced the precision medicine initiative in 2015, which was aimed at collecting comprehensive medical data from one million people. The data will be utilized in understanding disease pathogenesis, identifying the vulnerable population, and developing new therapeutic strategies. Breast cancer is one of the important diseases that is being studied for precision medicine.

Treatment regimen based on the molecular profiling of estrogen, progesterone, and HER2 receptors is a common way to treat breast cancer. However, this approach does not individualize a treatment strategy suitable for everyone. Currently, there are several commercially available gene profiling tests that are employed to assess the disease-free survival, risk of relapse, need for an adjuvant treatment, etc., based on gene profiling patterns. For instance, Oncotype DX (Genomic Health, CA, USA) analyzes the gene expression profiles of twelve genes that are recognized by the physician to determine if the patient requires adjuvant radiotherapy [309]. Mutations in PI3K-AKT-mTOR axis were correlated with the incidence of hormone receptor positive breast cancer [310]. 

miRNA profiling of serum is applied as a non-invasive liquid biopsy technique for the identification of breast cancer subtypes [311,312]. Next generation sequencing (NGS) is a deep sequence technique that, in parallel, sequences millions of DNA fragments to identify signature mutations in a single day. Mutations in TP53, PIK3CA, and GATA3 are the most common ones to monitor breast cancer progression [313]. Proteomic profiling of breast cancer tissues and cell lines have identified certain clusters that have a higher dysregulation of specific proteins [314]. Genome-wide association studies have identified fibroblast growth factor receptor 2 (FGFR2) and TOX high mobility group box family member 3 (TOX3) genes as highly associative of breast cancer [315,316]. Recently, 8q24.21 was identified as one of the loci to be associated with breast cancer [317]. Gene profiling is also applied to identify the adverse drug reactions and predict the drug response. For instance, screening for CYP2D6 for polymorphisms is associated with a shorter recurrence free survival in patients after tamoxifen treatment [318]. These studies indicate the potential bid for genomic data to stratify individuals for formulation of different treatment strategies and advancement.

## 6. Conclusions

Mechanisms of drug resistance in cancer are very complex. Molecular amendments, tumor microenvironment (hypoxic conditions leading to vascularization), and genetic rewriting are chief architects to cellular repair mechanisms, such as activation of DNA repair, mutant p53, impaired apoptosis, and so on. From pharmaceutical stand points, drug bioavailability at therapeutic concentrations in the target cancer cells is the major impediment of chemotherapy. Despite many recent discoveries on therapeutic interventions, cancer chemotherapy represents the most common treatment modalities for the disease. Successful treatment modalities impose three to four drugs at once. Patients in early stage of cancer respond, but due to MDR, cancer cells adapt treatment resistance and cause a relapse by overpowering therapy [319,320,321,322]. As a result of high and complex dosing regimens with multiple drugs, high toxicity is caused and many treatments fail. Hopefully, current trends on targeted nano-therapeutics may overcome some of these inadequacies and advent successful breast cancer therapy.

## Figures and Tables

**Figure 1 ijms-22-04673-f001:**
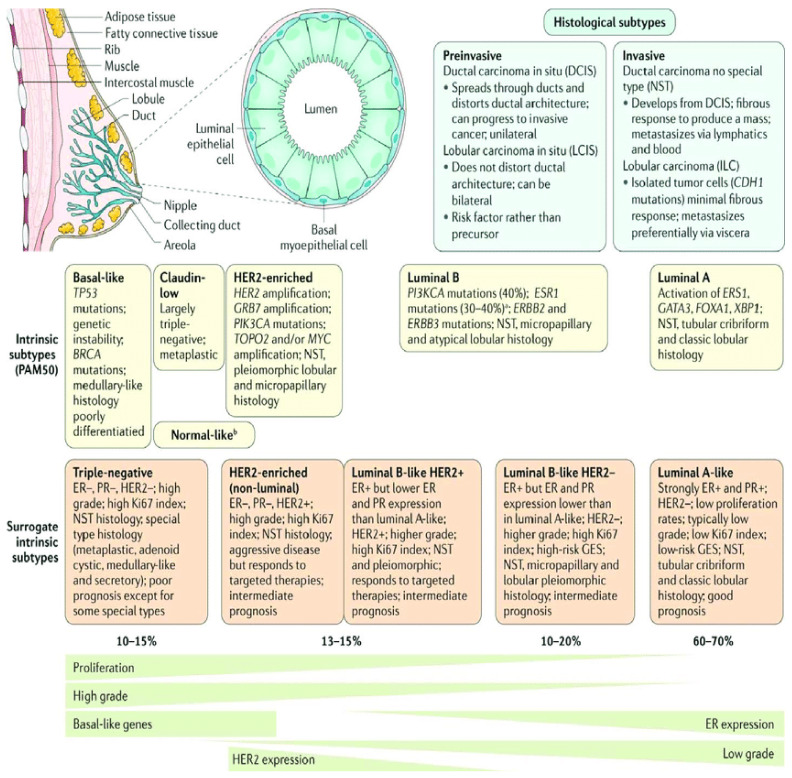
Breast cancer classification [2]. Histological and molecular characteristics of breast cancer have important implications for cancer therapy. Hence, several classifications based on molecular and histological characteristics are developed. The histological subtypes of breast cancer (top right) are the most frequent subtypes, including ductal carcinoma and lobular carcinoma, which are the invasive lesions, while their prevalent counterparts are ductal carcinoma in situ and lobular carcinoma in situ. Intrinsic subtype classification by Perou and Sorlie is based on a 50-gene expression signature (PAM50). The surrogate intrinsic subtype classification of breast cancer is used clinically, and it is based on immunohistochemistry and histology expression of key proteins, including progesterone receptor (PR), estrogen receptor (ER), human epidermal growth factor receptor 2 (HER2), as well as proliferation marker Ki67. Breast cancer tumors expressing ER and/or PR are “hormone receptor-positive”. Tumors lacking ER, PR but showing expression of HER2 are termed as “HER2 positive breast cancer”, while tumors lacking ER, PR, and HER2 are called “triple- negative breast cancer” [2].

**Figure 2 ijms-22-04673-f002:**
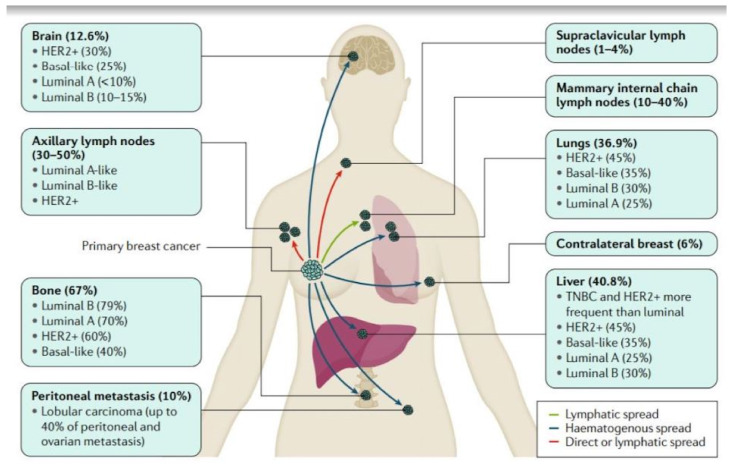
Illustration of common metastatic sites in breast cancer [2]. The most common metastatic sites for breast cancer are the bones, axillary lymph nodes, liver, and lungs. Approximately 10–40% of breast cancer tumors have metastasized in the internal mammary gland. Breast cancer can advance to distant metastatic sites depending on the molecular subtype; for example, Luminal A, B, HER2+, triple negative breast cancer (TNBC), or basal-like breast cancer. Locoregional lymphatic spread is sparse in TNBC compared to other subtypes, while brain metastases are more frequent in TNBC compared to luminal tumors [2].

**Figure 3 ijms-22-04673-f003:**
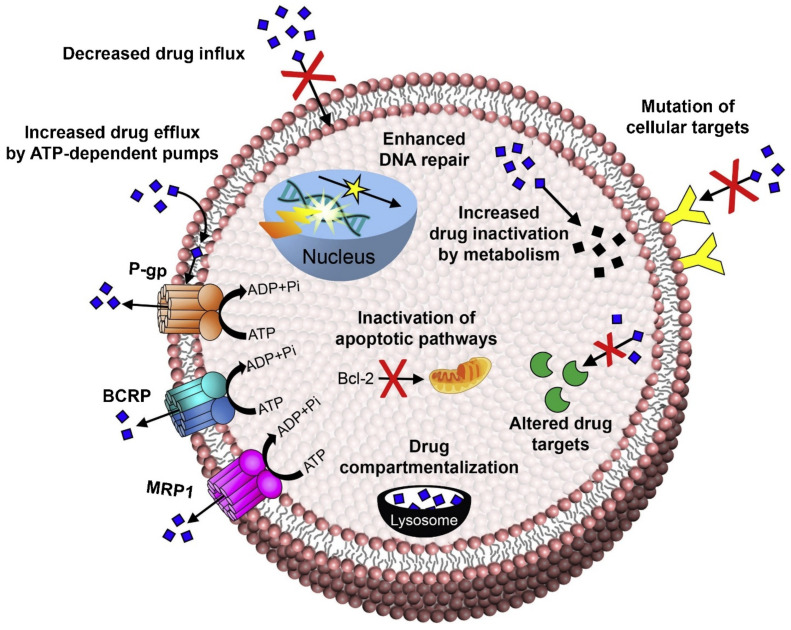
Mechanisms of drug resistance, including efflux pump-mediated mechanisms of MDR and efflux pump-independent drug resistance mechanisms [25]. The figure depicts how efflux transporters, including P-gp, BCRP, and MRP-1, efflux drugs and other xenobiotics from the cells. Cancer cells can develop drug resistance by increasing the expression of drug metabolizing enzymes and, hence, inactivating the drugs. Similarly, resistance can be developed by the cells by altering drug targets, rendering the drugs inactive. Drug compartmentalization into lysosomes causes drug inactivation and acquired drug resistance by cancer cells [25].

**Figure 4 ijms-22-04673-f004:**
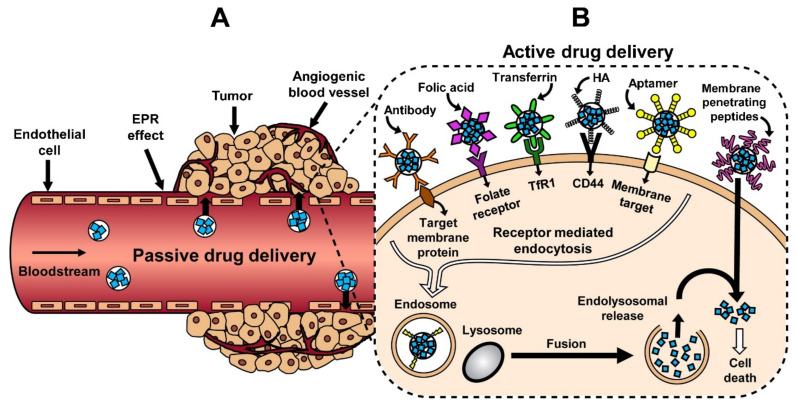
Drug targeting strategies: (**A**) passive drug delivery by the enhanced permeation and retention (EPR) effect and (**B**) active drug delivery mechanisms [25].

**Figure 5 ijms-22-04673-f005:**
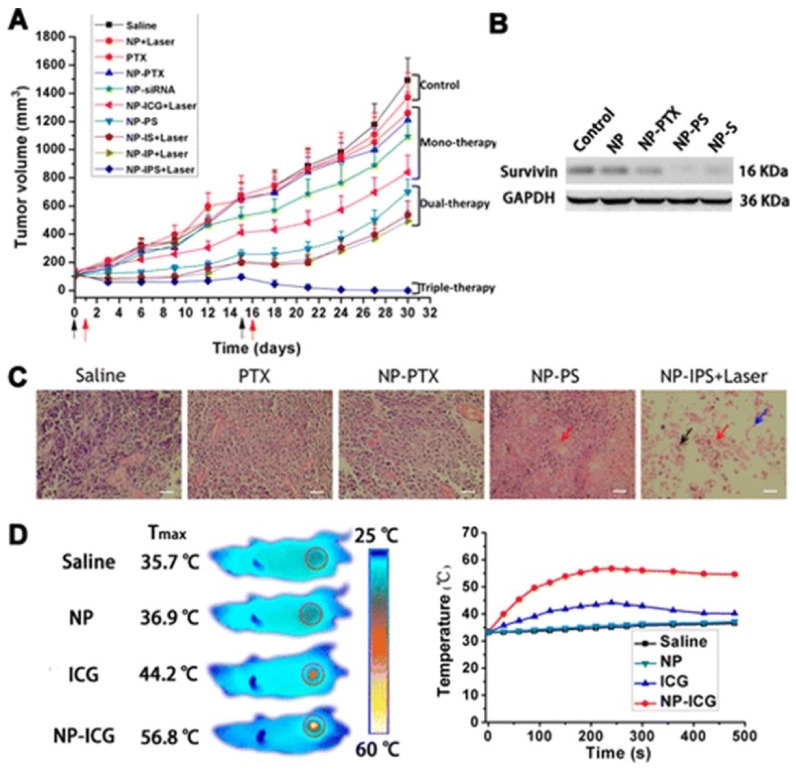
In vivo antitumor activity of NPs. (**A**) Antitumor activities of various drug formulations in a xenograft model of human breast cancer in BALB/c athymic nude mice. Mice bearing MDA-MB-231 tumors (~0.1 cm^3^) were treated with different reagents (*n* = 5). The treatment schedule was indicated by black arrows for intravenous injection and red arrows for the laser irradiation. (**B**) The expression of survivin in tumor tissues detected by Western blot 2 days after the second injection. (**C**) Representative H&E sections of tumors after treatment with saline, PTX, NP-PTX, NP-PS, or NP-IPS + Laser. Red arrow, karyolysis; blue arrow, abundant pykonosis; black arrow, coagulative necrosis. The tissue sections were 5 μm thick. Scale bars, 50 μm. (**D**) Infrared thermographic maps of tumors after laser irradiation for 8 min and maximum temperature profiles of the irradiated tumor areas of nude mice treated with NP-ICG, ICG, NP, or saline [237].

**Figure 6 ijms-22-04673-f006:**
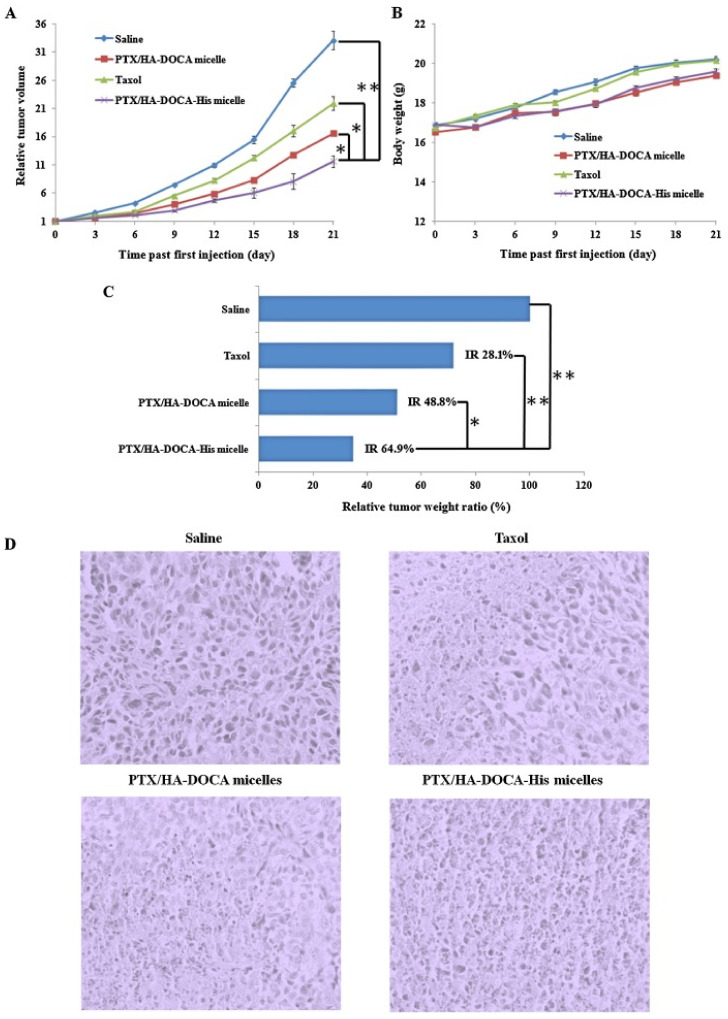
In vivo antitumor activity of Taxol, PTX/HA-DOCA, and PTX/HA-DOCA-His micelles in a multidrug-resistant human breast carcinoma MCF-7/Adr tumor model with saline as a control group. (**A**) Relative tumor volume changes of different treatments with time past first injection (mean ± SE, *n* = 5). * *p* < 0.05, ** *p* < 0.01 (PTX/HA-DOCA-His micelles vs. other treatment). (**B**) Changes of body weight in mice following different treatments. (**C**) Measured tumor weights after excision, plotted as relative tumor weight ratio. Tumor growth inhibition rate (IR, %) was calculated as: (1- (mean tumor weight of drug treated group/mean tumor weight of saline treated group)) × 100. (**D**) Histological analyses of tumor tissues treated with saline, Taxol solution, PTX/HA-DOCA, and PTX/HA-DOCA-His micelles collected on day 20 using H&E staining [243].

**Figure 7 ijms-22-04673-f007:**
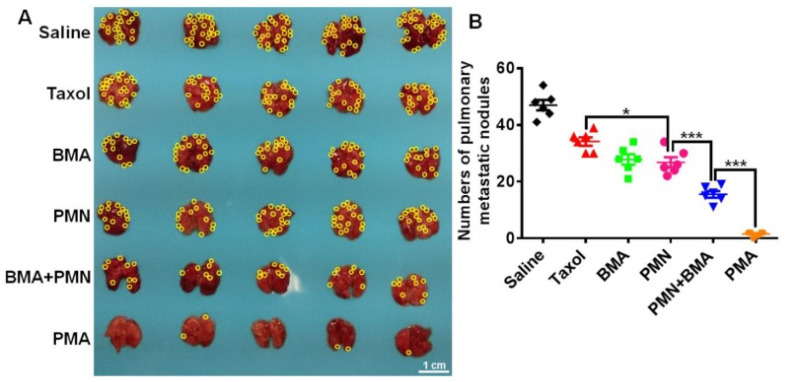
Anti-metastasis effect of different formulations in 4T1 tumor-bearing mice. (**A**) Images of the lungs at day 25. (**B**) Quantitative analysis of the pulmonary metastatic nodules at day 25. The yellow circles indicate metastatic nodules on the lungs. (Scale bar: 1 cm) Data are shown as mean ± SD (*n* = 5). * *p* < 0.05 and *** *p*< 0.001 [244].

**Figure 8 ijms-22-04673-f008:**
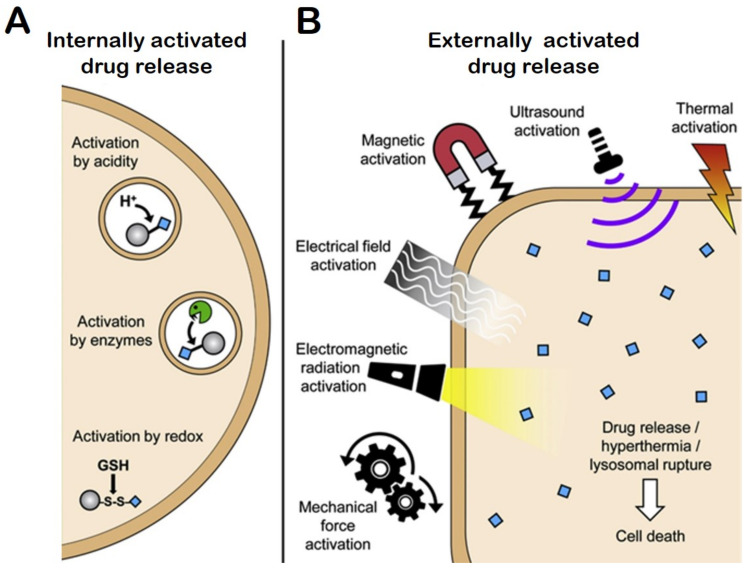
Drug triggering and release mechanisms of internally-activated (**A**) and externally-activated (**B**) drug release [25]. Stimuli sensitive drug release can be classified as internal-activated drug release and external activated drug release. Mechanisms including (i) activation of drug by change in pH, (ii) activation of drug by intracellular cancer enzymes, and (iii) redox-activated drug release are some of the prominent mechanisms for internal target-activated drug release. Some mechanisms most commonly used in research studies for external stimuli-responsive are (i) thermal activation, (ii) ultrasound activation, (iii) magnetic field activation (ultrasound mediated activation), (iv) electromagnetic radiation activation, and (v) magnetic force activation. These can aid or cause drug release of chemotherapeutic agents near the tumor site [25].

**Figure 9 ijms-22-04673-f009:**
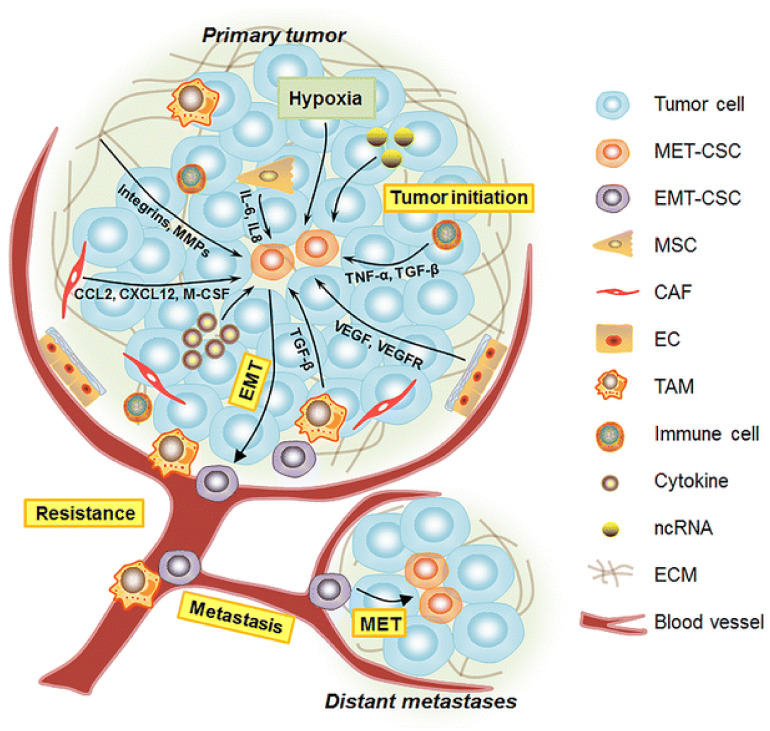
Breast cancer stem cell: the roles and therapeutic implications. Properties and regulation of BCSCs. This schematic diagram represents the interactions between CSCs and the surrounding tumor microenvironment, which have a direct effect on breast cancer cell malignancy and lead to tumor initiation, EMT, MET, metastasis, and therapeutic resistance [294].

**Table 1 ijms-22-04673-t001:** Passive drug targeting-based nanomedicines currently undergoing evaluation [25].

Nanomedicines	Chemotherapy Drug Payload	Overcoming Drug Resistance by	Tumor Type	Study Type	Reference
Cationic liposomes	Paclitaxel	MRP1 siRNA	Bcap-37—human breast cancer	In vitro	[117]
Cationic polymeric nanoparticles	Doxorubicin	MDR1 siRNA	MCF-7—human breast cancer expressing P-gp	In vitro	[118]
Nanomicellar amphiphilic dendrimer	Doxorubicin	Bypassing ATP-efflux pumps	MCF-7R—DOX-resistant human breast cancer	In vivo	[119]
Non-ionic surfactant-based vesicle (niosome)	Doxorubicin	Bcl-2 and ABCG2 siRNAs	MDA-MB-231—human breast cancer 231-CSCs human breast cancer stem	In vivo	[120]
PDPA/TPGS micelles	Doxorubicin	Vitamin E derivate (TPGS)	MCF7/ADR—DOX-resistant human breast cancer	In vivo	[121]
PEG-PBC micelles	Doxorubicin	Lapatinib	MCF7/ADR—DOX-resistant human breast cancer	In vivo	[122]
PLGA nanoparticles	Doxorubicin	Protamine − cell-penetrating peptide	MCF7/ADR—DOX-resistant human breast cancer	In vivo	[123]
Polypeptide cationic micelles	Docetaxel	Bcl-2 siRNA	MCF-7—human breast cancer overexpressing Bcl-2	In vivo	[124]
TPGS containing nanoemulsion	Paclitaxel	Vitamin E derivate (TPGS)	MCF-7/ADR—human breast cancer overexpressing P-gp	In vivo	[125]

**Table 2 ijms-22-04673-t002:** Recent progress in overcoming cancer drug resistance in breast cancer by using actively-targeted nanomedicines [25].

Nanomedicine	Ligand (Target)	Payload	Tumor type	Study Type	Reference
Eudragit RL100 nanoparticles	Hyaluronic acid (CD44 receptor)	Mitoxantrone Quercetin and Hesperetin	MCF-7, A2780p and A2780/ADR—human breast, ovarian and resistant ovarian cancer	In vitro	[149]
Cationic star-block terpolymer	Folic acid (Folate receptor)	Doxorubicin Bcl-2 siRNA	MCF-7—human breast cancer	In vitro	[150]
Lys-LA conjugates	Hyaluronic acid (CD44 receptor)	Doxorubicin	MCF-7/ADR—DOX-resistant human breast cancer	In vivo	[151]
Nanoparticles	IF7 peptide (Annexin 1)	Paclitaxel	MCF-7/ADR—multidrug-resistant human breast cancer	In vivo	[152]
PEG-PLA/PHIS-PEG pH sensitive micelles	Folic acid (Folate receptor)	Doxorubicin	MCF-7/ADR—multidrug-resistant human breast-cancer	In vivo	[153]
PEG-b-PGAH-b-PEI nanomicelleplexes	Folic acid (Folate receptor)	Doxorubicin MDR1 siRNA	MCF-7/ADR—multidrug-resistant human breast cancer	In vivo	[154]
α-TOS-TPGS	Hyaluronic acid (CD44 receptor)	Docetaxel	MCF-7/ADR—multidrug-resistant human breast cancer	In vivo	[155]
PLGA-PEG nanoparticles	Folate (Folate receptor)	Docetaxel PI3 K/Akt inhibitor	MCF-7/ADR—multidrug-resistant human breast cancer	In vitro	[156]
Pluronic pH and redox responsive micelles	Folic acid (Folate receptor)	Doxorubicin	MCF-7/MDR—multidrug-resistant human breast cancer	In vivo	[157]
Core-shell nanomicelles	Folic acid (Folate receptor)	Doxorubicin MDR1 siRNA	MCF-7/ADR—DOX-resistant human breast cancer	In vivo	[158]
PCDA-PEG nanoparticles	Biotin (Biotin receptor)	Doxorubicin Curcumin	MCF-7/ADR—multidrug-resistant human breast cancer	In vivo	[159]

**Table 3 ijms-22-04673-t003:** Breast Cancer Stem Cells (BCSCs) targeting therapies in the clinical trials [294].

Agent	Target	Trial Phase (Trial Number, Status)	Patients (Number)	Combined Therapy
Notch pathway-targeting				
MK-0752	γ-secretase	Phase I (NCT00756717, active)	Early stage, ER-positive breast cancer (22)	Tamoxifen or AI
		Phase I/II (NCT00645333, completed)	Advanced or metastatic breast cancer (30)	Docetaxel
		Phase I (NCT00106145, completed)	Metastatic or locally advanced breast cancer (24) and other solid tumors (79)	NA
RO4929097	γ-secretase	Phase I (NCT01238133, terminated)	TNBC (14)	Paclitaxel and carboplatin
		Phase I (NCT01071564, terminated)	Advanced or unresectable breast cancer (13)	Vismodegib
		Phase I (NCT01149356, terminated)	Advanced or metastatic breast cancer (15)	Exemestane
		Phase I (NCT01208441, terminated)	Post-menopausal hormone receptor-positive stage II/III breast cancer (28)	Letrozole
		Phase II (NCT01151449, active)	Advanced, metastatic, or recurrent TNBC (3)	NA
PF-03084014	γ-secretase	Phase II (NCT02299635, terminated)	Advanced TNBC (19)	NA
		Phase I (NCT01876251, terminated)	Advanced breast cancer (30)	Docetaxel
Hedgehog pathway-targeting				
GDC-0449 (vismodegib)	Smoothened	Phase I (NCT01071564, terminated)	Metastatic or unresectable breast cancer (13)	RO4929097
		Phase II (NCT02694224, recruiting)	TNBC (40)	Paclitaxel, epirubicin, and cyclophosphamide
Wnt pathway-targeting				
OMP-18R5 (vantictumab)	Frizzled7	Phase I (NCT01973309, recruiting)	Locally recurrent or metastatic breast cancer (34)	Paclitaxel
BCSC-targeting				
Bivatuzumab mertansine	CD44v6	Phase I (NCT02254005, completed)	CD44v6-positive metastatic breast cancer (24)	NA
		Phase I (NCT02254031, terminated)	CD44v6-positive recurrent or metastatic breast cancer (8)	NA
CSC vaccine	BCSC	Phase I/II (NCT02063893, completed)	(completed) metastatic breast cancer (40)	NA
Multiplasmid vaccine	CD105/Yb-1/SOX2/CDH3/MDM2	Phase I (NCT02157051, recruiting)	HER2-negative stage III/IV breast cancer (30)	NA
Microenvironment-targeting				
Reparixin	CXCR1/2	Phase I (NCT02001974, completed)	HER2-negative metastatic breast cancer (33)	Paclitaxel
		Phase II (NCT02370238, recruiting)	Metastatic TNBC (190)	Paclitaxel
		Phase II (NCT01861054, terminated)	Early breast cancer (20)	NA

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
