# Peer review of "Drug Resistance in Metastatic Breast Cancer: Tumor Targeted Nanomedicine to the Rescue"

_ijms, 2021, doi:10.3390/ijms22094673_

Round 1

Reviewer 1 Report

The review article “Drug Resistance in Metastatic Breast Cancer: Tumor Targeted Nanomedicine to the rescue” written by Gote and colleagues provides a comprehensive overview about the current trends in nanomedicine. The referred literature is broad and mostly well presented.

A couple of minor issues should be considered by the authors:

1. The reference to figures 3 is not present in the main text.

2. Most of the figure legends are to scarce, especially Figure 1, 2, 3 and 8. A more profound description of the shown schematic drawings would be necessary for the readers to fully understand the content of the figures.

3. In the section1.2 “Breast Cancer Pathophysiology and Metastasis” the aspect metastasis should be more prominent. Although figure 2 is provided, the connection between the described molecular mechanisms and metastasis is not clear. Figure 2 should be better integrated.

4. At several positions, sentences are difficult to understand or somehow truncated and should be revised (e.g. Page 4, line 125 or Page 5 line 147 – 149)

5. Authors refer to Figure 8B; however these Figure panel is not presented.

Author Response

The review article “Drug Resistance in Metastatic Breast Cancer: Tumor Targeted Nanomedicine to the rescue” written by Gote and colleagues provides a comprehensive overview about the current trends in nanomedicine. The referred literature is broad and mostly well presented.

The authors would like to thank the reviewer for their thoughtful and careful review of the manuscript. Certain corrections like proper explanations for the figures and better explanation of the figures in the text have definitely improved the quality of the review. We are thankful for the time and efforts the reviewer has put for this review. We also hope that the reviewer finds the changes acceptable.

A couple of minor issues should be considered by the authors:

  1. The reference to figures 3 is not present in the main text.

Figure 3 is now mentioned in the main text.

  1. Most of the figure legends are to scarce, especially Figure 1, 2, 3 and 8. A more profound description of the shown schematic drawings would be necessary for the readers to fully understand the content of the figures.

The legends for Figure 1, 2, 3 and 8 are now more elaborate explain the figures in more detail.

  1. In the section1.2 “Breast Cancer Pathophysiology and Metastasis” the aspect metastasis should be more prominent. Although figure 2 is provided, the connection between the described molecular mechanisms and metastasis is not clear. Figure 2 should be better integrated.

Figure 2 is now deeply explained at the end of the section 1.2. A new paragraph describing the various sites of breast cancer metastasis is now added in the revised manuscript. The various aspects of metastasis are explained in the section additionally. This explains the mechanisms involved and the pathways of metastasis.

  1. At several positions, sentences are difficult to understand or somehow truncated and should be revised (e.g. Page 4, line 125 or Page 5 line 147 – 149)

This is now revised in the updated manuscript.

  1. Authors refer to Figure 8B; however these Figure panel is not presented.

This is now revised in the updated manuscript.

Reviewer 2 Report

Manuscript ID: ijms-1182852

Type of manuscript: Review

Title: Drug Resistance in Metastatic Breast Cancer: Tumor Targeted Nanomedicine to the rescue

Authors: Vrinda Gote, Anantha Ram Nookala, Pradeep Kumar Bolla, Dhananjay Pal*

The manuscript Gote et al. describes very thoroughly various nanomedicines designed to enhance drug uptake, discussed recent approaches in possible treatment of breast cancer such as combination of nanoparticles, theranostic, intrinsic and extrinsic stimuli, sensitive nanoparticles, microRNA and personalized medicine nanoparticles for selective accumulation, increased efficacy and MDR reversal in breast cancer.   This manuscript seems to be well written, detailed. The most of references are new, up to date and relevant.  Before the manuscript is accepted by IJMS I would like to suggest authors to include following references in the text as well since they are describing similar topics and it will be nice to see how this manuscript differ from these ones:

  • Nanoparticle delivery of anticancer drugs overcomes multidrug resistance in breast cancer, Yueling Yuan, Tiange Cai, Xi Xia, Ronghua Zhang, Peter Chiba & Yu Cai, 2016, Delivery Volume 23 (9), 3350-3357
  • Nanoparticle-Based Drug Delivery in Cancer Therapy and Its Role in Overcoming Drug Resistance, Yihan Yao, Yunxiang Zhou, Lihong Liu, Yanyan Xu, Qiang Chen, Yali Wang, Shijie Wu, Yongchuan Deng, Jianmin Zhang, and Anwen Shao, Front Mol Biosci. 2020; 7: 193.

  • Chemoresistance mechanisms of breast cancer and their countermeasures, Biomedicine & Pharmacotherapy, Volume 114, June 2019, 108800
  • Application of nanoparticles to reverse multi-drug resistance in cancer, Jie Yang , Haijun Zhang and Baoan Chen, De Gruyter | Published online: July 13, 2016, DOI: https://doi.org/10.1515/ntrev-2016-0023

Further:

  • Avoid personalization in abstract: Line 25, 27 and 31
  • Remove dot in Line 243 between …mechanism and (25)
  • The sentence Line 472-475 is not clear since it is known that DOX is actually inducer of oxidative damages.
  • Line 550 correct Taxus Bucata in Taxus baccata and European tree in European yew tree. Also, since the authors described here specifically origin of docetaxel the same should be performed for paclitaxel (isolated from Taxus brevifolia (Pacific yew)).
  • Remove underline under 242 (Line 746)
  • The sentence Line 814-815 is confusing.
  • Line 820, correct In vivo in In vivo
  • The sentence Line 935-937 is not clear.
  • Table 3…correct the style of column 3 (comma issue)
  • The font of all tables is too big.
  • Figures are not uniform in style.

Author Response

The manuscript Gote et al. describes very thoroughly various nanomedicines designed to enhance drug uptake, discussed recent approaches in possible treatment of breast cancer such as combination of nanoparticles, theranostic, intrinsic and extrinsic stimuli, sensitive nanoparticles, microRNA and personalized medicine nanoparticles for selective accumulation, increased efficacy and MDR reversal in breast cancer.   This manuscript seems to be well written, detailed. The most of references are new, up to date and relevant.  Before the manuscript is accepted by IJMS I would like to suggest authors to include following references in the text as well since they are describing similar topics and it will be nice to see how this manuscript differ from these ones:

The authors thank the reviewer for their thoughtful and careful review of the manuscript. The reviewers comments have definitely improved the quality of the review. We are thankful for the time and efforts the reviewer has put for this review. We also hope that the reviewer finds the changes acceptable.

Nanoparticle delivery of anticancer drugs overcomes multidrug resistance in breast cancer, Yueling Yuan, Tiange Cai, Xi Xia, Ronghua Zhang, Peter Chiba & Yu Cai, 2016, Delivery Volume 23 (9), 3350-3357

Nanoparticle-Based Drug Delivery in Cancer Therapy and Its Role in Overcoming Drug Resistance, Yihan Yao, Yunxiang Zhou, Lihong Liu, Yanyan Xu, Qiang Chen, Yali Wang, Shijie Wu, Yongchuan Deng, Jianmin Zhang, and Anwen Shao, Front Mol Biosci. 2020; 7: 193.

Chemoresistance mechanisms of breast cancer and their countermeasures, Biomedicine & Pharmacotherapy, Volume 114, June 2019, 108800

Application of nanoparticles to reverse multi-drug resistance in cancer, Jie Yang , Haijun Zhang and Baoan Chen, De Gruyter | Published online: July 13, 2016, DOI: https://doi.org/10.1515/ntrev-2016-0023

 These references are now added in the conclusion section.

Further:

Avoid personalization in abstract: Line 25, 27 and 31

This is now corrected in the revised manuscript.

Remove dot in Line 243 between …mechanism and (25)

This is now corrected in the revised manuscript

The sentence Line 472-475 is not clear since it is known that DOX is actually inducer of oxidative damages.

This is now corrected in the revised manuscript.

Line 550 correct Taxus Bucata in Taxus baccata and European tree in European yew tree. Also, since the authors described here specifically origin of docetaxel the same should be performed for paclitaxel (isolated from Taxus brevifolia (Pacific yew)).

This is now corrected in the revised manuscript

Remove underline under 242 (Line 746)

This is now corrected in the revised manuscript

The sentence Line 814-815 is confusing.

This is now corrected in the revised manuscript

Line 820, correct In vivo in In vivo

This is now corrected in the revised manuscript

The sentence Line 935-937 is not clear.

This is now corrected in the revised manuscript

Table 3…correct the style of column 3 (comma issue)

This is now corrected in the revised manuscript

The font of all tables is too big.

This is now corrected in the revised manuscript. The font is reduced and a same style is maintained throughout.

Figures are not uniform in style.

This is now corrected in the revised manuscript